# Subgroup Discovery with the Cox Model

**Zachary Izzo** [1]   **Iain Melvin** [1]

## Abstract

We study the problem of subgroup discovery for survival analysis, where the goal is to find an interpretable subset of the data on which a Cox model is highly accurate. We examine why existing quality functions are insufficient for this problem and introduce two technical innovations: the *expected prediction entropy (EPE)*, a novel metric for evaluating survival models that predict hazard functions, and the *conditional rank statistics (CRS)*, which quantifies individual point deviation from a subgroup's survival time distribution. We study the EPE and CRS theoretically and show they address problems with existing metrics. We then introduce eight algorithms for Cox subgroup discovery. Our main algorithm is based on the DDGroup framework of Izzo et al. (2023) and leverages both the EPE and CRS, allowing theoretical correctness guarantees in well-specified settings. Empirical evaluation on synthetic and real data confirms our theory, showing our methods recover ground-truth subgroups in well-specified cases and achieve better model fit than naively fitting the Cox model to the entire dataset. A case study on NASA jet engine simulation data demonstrates that discovered subgroups uncover known nonlinearities in the data and suggest design choices mirrored in practice.

## 1. Introduction

The Cox model (Cox, 1972; 1975) is widely used in fields ranging from biostatistics to manufacturing (Hosmer Jr et al., 2008; Kalbfleisch & Prentice, 2011), both for prediction (Huo et al., 2024) and for qualitative inference about the data (Liu et al., 2022; 2024b). While the Cox model is appealing for its ease of interpretation, it makes restrictive modeling assumptions which are known not to hold in some practical

scenarios (Hernán, 2010). Other methods for survival analysis, including many deep learning-based approaches, have been proposed by the ML community (Wang et al., 2019). These models are more expressive and therefore do not require making restrictive assumptions about the data, but this comes at the cost of interpretability. Especially in high-stakes settings, or when we want to use the model not only for prediction but for qualitative inference about the data, this may be unacceptable (Rudin, 2019). Thus, it is valuable to develop methods which improve the modeling capacity of the Cox model without sacrificing interpretability.

To fill this gap, we use a subgroup discovery-based approach (Lipkovich et al., 2023). The idea is that while the Cox model may not be a good fit for the entire dataset, there may be local subpopulations of the data (referred to as *subgroups*) where the Cox model is appropriate. In order to preserve the overall interpretability of the method, it is desirable that the subgroup descriptions also be interpretable. We follow prior work (Izzo et al., 2023) and define the subgroups by thresholding different feature values. The resulting subgroups correspond to axis-aligned boxes in feature space. The interpretable subgroup definitions mean that the subgroups themselves may be useful in defining meaningful subpopulations with homogeneous, predictable outcomes, which may be useful for follow-up studies. In short, rather than sacrificing accuracy or interpretability, we sacrifice our ability to model the entire population simultaneously, and our goal can be summarized as:

*Find interpretable subsets of the data in which a Cox model is an excellent fit.*

In this paper, we address several fundamental questions for the problem of subgroup discovery with the Cox model. We first look at the barriers to solving the problem using existing techniques. Specifically, subgroup discovery methods generally rely on a "quality function" to evaluate potential subgroups, allowing these algorithms to sift through and select a favorable subgroup. In the context of the Cox model, two natural choices for quality functions are Harrell's C-index (Harrell et al., 1982) and the Cox partial likelihood (Cox, 1975). By means of counterexamples and empirical evaluations, we show that both of these metrics have undesirable properties which make them unsuitable for solving the Cox subgroup discovery problem.

---

[1]Machine Learning Department, NEC Labs America. Correspondence to: Zachary Izzo <zach@nec-labs.com>.

*Proceedings of the 43rd International Conference on Machine Learning*, Seoul, South Korea. PMLR 306, 2026. Copyright 2026 by the author(s).

To address the shortcomings of existing metrics, we make two novel technical contributions: the *expected prediction entropy (EPE)*, an alternative metric for the performance of a Cox model (or any survival model which estimates a hazard function); and the *conditional rank statistics (CRS)*, which measures the plausibility that a test point follows the same Cox model as some reference group of points. We study the theoretical properties of both of these quantities and show that they address some of the shortcomings of the existing metrics considered earlier.

Having established fundamental results on the Cox subgroup discovery problem, we next develop methodology for solving it. Indeed, the EPE and CRS, in addition to being motivated by the shortcomings of the C-index and partial likelihood, are also motivated by the general subgroup discovery framework of Izzo et al. (2023). The idea is to first select a small "core group" of points with a good fit to the Cox model; this stage is accomplished with the EPE. Next, we examine the remaining points in the dataset and "reject" those which could not feasibly follow the same model as the core group; in the case of survival analysis, this can be accomplished with the CRS. Finally, we define the subgroup by expanding the core group as much as possible without including any rejected points. By taking advantage of both the EPE and CRS, we prove that this "survival analog" of DDGroup can provably recover a ground truth subgroup in a well-specified setting.

To the best of our knowledge, we are the first to consider this formulation of the subgroup discovery problem with the Cox model. As a result, there are no existing baseline methods, so in addition to our primary algorithm, we also introduce eight other methods for comparison. We then evaluate all eight algorithms (DDGroup + seven baselines) against the non-subgroup discovery approach of simply fitting a Cox model to the entire dataset on both real and synthetic data. These experiments confirm our theory and show the practical utility of the proposed methods. In addition to evaluating prediction performance, we also conduct a case study on simulated jet engine failure data from NASA (Saxena et al., 2008) to show the value of the subgroups themselves. We are able to recover subgroups corresponding to known non-linearities in the data and which suggest follow-up studies and design choices which have indeed been mirrored in practice, showing the value of these subgroup methods for hypothesis generation.

**Summary of Contributions**

1. We analyze the shortcomings of existing metrics, such as the C-index and partial likelihood, for solving the problem of subgroup discovery with the Cox model.

2. We introduce the *expected prediction entropy (EPE)* as an evaluation metric for survival model accuracy. We derive

several properties of the EPE to assist in its interpretation as an evaluation metric.

3. We introduce the *conditional rank statistics (CRS)*, which quantifies the deviation of a test point to the distribution of survival times in an existing subgroup.

4. We introduce eight algorithms for subgroup discovery with the Cox model. We establish theoretical correctness guarantees for our main algorithm which makes use of both the EPE and the CRS.

5. We apply these algorithms to both real and synthetic datasets and analyze the subgroups discovered by each method.[1] We also conduct a case study to demonstrate the utility of discovered subgroups for hypothesis generation as well as improved predictive power.

**Related Work** The Cox model (Cox, 1972; 1975) is a standard method for survival analysis widely used in practice due to its interpretability, though this comes at the cost of strong modeling assumptions that may be violated (Hernán, 2010). The machine learning community has developed numerous modern techniques to provide more flexible survival models (Katzman et al., 2018; Hu & Nan, 2023; Wu et al., 2023; Bleistein et al., 2024; Bertsimas et al., 2022; Zhang et al., 2024; Huisman et al., 2024; Kim, 2023; Chen, 2024; Lee et al., 2018; Rindt et al., 2022; Vauvelle et al., 2023; Wang et al., 2019; Ching et al., 2018; Che et al., 2018; Ripley, 1998; Giunchiglia et al., 2018; Liu et al., 2024a). Harrell's concordance index (C-index) (Harrell et al., 1982) is the most common evaluation metric for survival models, evaluating how well predicted failure order matches observed data, though it has many known shortcomings (Hartman et al., 2023) that motivate recent interest in proper evaluation metrics (Yanagisawa, 2023; Haider et al., 2020; Qi et al., 2023). Our work lies at the intersection of survival analysis and subgroup discovery, which broadly refers to mining datasets for regions where the data distribution is "interesting" according to some numerical score function (Friedman & Fisher, 1999; Atzmueller, 2015; Leman et al., 2008; Izzo et al., 2023; Xu et al., 2024). Subgroup discovery has found extensive applications in biostatistics (Lipkovich et al., 2017a; 2023), particularly for studying heterogeneous treatment effects to identify patient groups experiencing enhanced treatment benefit (Kehl & Ulm, 2006; Lipkovich et al., 2011; Dusseldorp & Van Mechelen, 2014; Lipkovich & Dmitrienko, 2014; Lipkovich et al., 2017b; Schnell et al., 2018; Schnell, 2021; Li & Imai, 2025) and for patient stratification (Polonik & Wang, 2010; Chen et al., 2015; Huang et al., 2017). The work most closely related to ours is Wei & Kosorok (2018), which studied subgroup discovery for the Cox model but defined subgroups via hyperplane sides

---

[1]Code for reproducing our experiments can be found at https://github.com/zleizzo/cox-subgroup.

rather than axis-aligned boxes. It also imposes more restrictive assumptions on subgroup relationships, rendering it unsuitable for our setting as discussed in Appendix H.1; Zhang et al. (2025) also studied properties of this change-plane Cox model.

**Conflict of Interest Disclosure**   ZI and IM are employed by NEC Laboratories America, Inc., which holds a patent to the proposed method.

## 2. Background and Notation

We briefly recap notation and basic facts about survival analysis and the Cox model. Readers unfamiliar with these topics can find more background in Appendix A.

In survival analysis, the goal is to model the distribution of a non-negative survival time $T$ (the time until a *unit* (i.e., some entity for which we collect a datapoint) experiences some event, called a *failure*) conditional on covariates $X \in \mathbb{R}^d$. Survival data will frequently be subject to *censoring*: rather than observing $T$ directly, there is another nonnegative random variable $C$ (the censoring time) and we only observe $\min(T, C)$ as well as the indicator $\delta = \mathbb{1}\{T \leq C\}$ of whether the failure occurred or the datapoint was censored. The primary modeling target is the hazard function $\lambda(t, x) = \lim_{dt \to 0} \frac{\mathbb{P}(t \leq T \leq t + dt \mid T \geq t, X = x)}{dt}$, which represents the instantaneous failure rate at time $t$ given survival to that point. The Cox model (Cox, 1972; 1975) posits a semiparametric form $\lambda(t, x) = \lambda_0(t) \exp(\beta^\top x)$ where $\beta$ are coefficients to be estimated; higher values of $\beta^\top x$ predict faster failure. Given failure times $t_1 \leq \ldots \leq t_n$, features $x_i$, and failure indicators $\delta_i$, the coefficients $\beta$ are estimated by maximizing the log partial likelihood $\mathcal{L}(\beta) := \sum_{1 \leq i \leq n, \, \delta_i = 1} \left[ x_i^\top \beta - \log \left( \sum_{j \geq i} \exp(x_j^\top \beta) \right) \right]$.

## 3. Expected Prediction Entropy

As discussed in the introduction, we are guided in part by the desire to adapt the DDGroup framework of Izzo et al. (2023) to the survival analysis setting. The first step of this framework is to select a small "core group" of points where the Cox model gives a good fit. In general, we also want precise metrics to evaluate the quality of discovered subgroups, especially in realistic settings where ground truth subgroup descriptions may not be known. In this section, we discuss the shortcomings of existing metrics which appear to be natural choices for these tasks, and introduce the expected prediction entropy (EPE) to address these shortcomings.

### 3.1. Inadequacy of Existing Metrics

One of the most common measures of model accuracy in survival analysis is Harrell's C-index (Harrell et al., 1982),

which measures the fraction of comparable units for which the earlier failure time coincides with the model's prediction. This metric is similar to the 0-1 loss in classification, and while it is a useful summary statistic, it is insensitive to the confidence of the model's prediction: a model prediction which was incorrect with 51% confidence is penalized equally to a prediction which was incorrect with 99% confidence. As a result, evaluation using only the C-index can obscure the existence of small subpopulations with qualitatively different behavior from the rest of the population. In the context of subgroup discovery, these subpopulations are exactly what we want to detect, making the C-index inadequate for this task.

A natural alternative to consider is the partial likelihood. While the partial likelihood does take model confidence into account, it is not suitable for *comparing* different groups of data. To see this, consider the following example. If the first unit to fail out of 1000 units was given a predicted 10% chance of being the first to fail by the model, this could be considered a very confident and accurate prediction (a $100\times$ improvement over a random guess, which would assign each unit a $1/1000$ chance of failure). On the other hand, if only two units were at risk and the model assigned a 10% chance of failure to the unit which failed first, this would constitute a confident but inaccurate prediction. However, these two scenarios contribute equally to the partial likelihood.

### 3.2. EPE Definition

Let $\lambda(t, x)$ be the true hazard function. Conditional on a failure occurring at time $t$ among two units with features $x_1$ and $x_2$, the probability that $x_1$ experiences failure is

$$\lambda(t, x_1) / (\lambda(t, x_1) + \lambda(t, x_2)). \tag{1}$$

Given a survival model which predicts a hazard rate $\hat{\lambda}(t, x)$, we can evaluate our model by measuring its ability to discriminate between which of two units at risk will fail.

**Definition 3.1** (Expected Prediction Entropy). Let $P$ be a probability distributions over $\mathbb{R}^d \times \mathbb{R}_{\geq 0}$ which denotes the joint distribution of a (feature, survival time) pair. Let $(X, T), (X', T') \sim P$ be two i.i.d. draws from $P$, let $T^* = \min\{T, T'\}$, and define $Y = \mathbb{1}\{T \leq T'\}$. Let $\hat{\lambda}$ be an estimate for the hazard function which defines the distribution of $T$ conditional on $X$, and let $R \subseteq \mathbb{R}^d$ be a sub-region of the feature space. We define the *expected prediction entropy (EPE)* as

$$\mathrm{EPE}(\hat{\lambda}, R) = \mathbb{E}\Bigg[ -Y \log \frac{\hat{\lambda}(T^*, X)}{\hat{\lambda}(T^*, X) + \hat{\lambda}(T^*, X')} \tag{2}$$
$$- (1 - Y) \log \frac{\hat{\lambda}(T^*, X')}{\hat{\lambda}(T^*, X) + \hat{\lambda}(T^*, X')} \,\Bigg|\, X, X' \in R \Bigg].$$

**Specialization to the Cox Model** When $\hat{\lambda}$ is given by a Cox model, i.e., $\hat{\lambda}(t, x) = \lambda_0(t)e^{\beta^\top x}$, (2) reduces to

$$\mathbb{E}\left[ -Y \log \frac{1}{1 + e^{-\beta^\top (X - X')}} \right. \tag{3}$$
$$\left. - (1 - Y) \log \frac{1}{1 + e^{\beta^\top (X - X')}} \mid X, X' \in R \right].$$

Observe that this is the standard cross entropy loss for a logistic model trained to predict the label $Y$ from the feature differences $X - X'$. We remark that the expression (3) appeared in Steck et al. (2007) as a lower bound for the C-index. The authors use this lower bound directly to train a Cox model, instead of the standard partial likelihood. Kvamme et al. (2019) used the same expression as an approximation to the partial likelihood, using a risk set of size 1 to avoid memory constraints during model training. Vauvelle et al. (2023) also explored this expression in the context of ranking losses, which are again used to train relative risk models. To the best of our knowledge, we are the first to explore the usefulness and properties of the EPE as an *evaluation metric*, not merely as a loss function.

**Estimating EPE Empirically** Let $\{(x_i, t_i, \delta_i)\}_{i=1}^n \subseteq \mathbb{R}^d \times \mathbb{R}_{\geq 0} \times \{0, 1\}$ be a survival dataset with features $x_i$, event times $t_i$, and censoring indicators $\delta_i$. An empirical estimate of the EPE is given by

$$-\frac{1}{N} \sum_{i : \delta_i = 1} \sum_{j \in R_i} \log \frac{\hat{\lambda}(t_i, x_i)}{\hat{\lambda}(t_i, x_i) + \hat{\lambda}(t_i, x_j)}, \tag{4}$$

where $R_i = \{j : t_j > t_i\}$ is the risk set at time $t_i$ (minus the $i$-th datapoint itself) and $N = \sum_{i : \delta_i = 1} |R_i|$ is the total number of comparable event times. In the case that there is no censoring (i.e., $\delta_i = 1$ for all $i$), (4) gives an unbiased estimate for (2). In the presence of censoring, the fact that we can only compare two datapoints when the first event time was uncensored may introduce a bias.

### 3.3. Properties of the EPE

In this subsection, we establish favorable theoretical properties of the EPE. We first show that when the data are generated by a Cox model, the EPE is a proper scoring rule in the sense that it is minimized if and only if the estimated Cox coefficients match the ground truth. This is the content of Proposition 3.2.

**Proposition 3.2.** *Suppose that the ground truth hazard function follows the Cox model, i.e., $\lambda(t, x) = \lambda_0(t)e^{\beta^\top x}$. Then the EPE is minimized iff $\hat{\beta}^\top (X - X') = \beta^\top (X - X')$ with probability 1.*

The proof is given in Appendix D.1. We remark that in the more general setting of (2), the EPE is also minimized by any scalar multiple of the ground truth (full) hazard function. This is unavoidable, since the EPE is relative in nature

and scaling the ground truth hazard will not change the probability of failure of one unit over another. However, multiplication of the hazard by a scalar also does not change the Cox coefficients, as this multiplication is absorbed into $\lambda_0(t)$, so the minimum EPE will uniquely identify the correct Cox coefficients. As the semiparametric nature of the Cox model means it is also inherently relative, this is the best we can hope for in terms of a proper scoring rule.

Next, we show that as long as the Cox model is well-specified, the EPE implicitly favors larger groups. This is beneficial in practice as we would like our conclusions to be applicable to as much of the data as possible.

**Proposition 3.3.** *Let the joint data distribution $P$ be such that the marginal distribution of the features $P|_X$ is uniform on a region $\mathcal{B} \subseteq \mathbb{R}^d$. Let $R = \prod_{i=1}^d [a_i, b_i]$, $R' = \prod_{i=1}^d [a_i', b_i']$ be axis-aligned boxes such that $R, R' \subseteq \mathcal{B}$ and such that $|a_i - b_i| \leq |a_i' - b_i'|$ for all $i$. Further suppose that $T|X$ follows a Cox model with coefficients $\beta$ whenever $X \in R \cup R'$. Then $\mathrm{EPE}(\beta, R') \leq \mathrm{EPE}(\beta, R)$.*

Because the minimum EPE depends on the intrinsic difficulty of distinguishing between units, we make the assumption that the features are uniformly distributed to remove this potential source of variation. To see why some assumption on the covariates is necessary, consider features which are drawn uniformly at random from $\mathcal{B}$ w.p. 1%, but with probability 99% they are equal to some fixed $x_0 \in \mathcal{B}$. As long as the region $R$ does not contain $x_0$, we obtain the same behavior as in the theorem, as the features will be conditionally uniformly distributed in $R$. However, once $R$ expands to include $x_0$, with high probability we will have $X = X'$ and the failure probability is 50-50, leading to a larger EPE. This counterexample can be smoothly approximated with purely continuous distributions.

In general, the practical interpretation of Theorem 3.3 is that provided that the Cox model is always well specified, the EPE favors a group where the intrinsic difficulty of distinguishing between units is the lowest; in the case of uniformly distributed features, larger groups mean that units will tend to be farther apart, making them easier to classify.

## 4. Conditional Rank Statistics

The EPE provides an improved metric for the fit of the Cox model to a group of points and can be used for the core group selection step in the DDGroup framework. Next, we turn to the second phase of DDGroup, where we must "reject" points which cannot feasibly follow the same model as the core group. For this task, we introduce the conditional rank statistics (CRS).

## 4.1. Motivation and Counterexamples

In the discussion of Theorem 3.3, we noted that the result implies that the EPE favors regions where units are intrinsically easier for the Cox model to distinguish, and this can be impacted not only by the hazard function but also the feature distribution. In particular, the EPE may prefer a region $R_1$ which does not follow a Cox model to a region $R_2$ which does follow a Cox model, provided that the units in $R_1$ are somehow "intrinsically easier" to distinguish between than the units in $R_2$. Figure 1 shows a heat map of the numerical value of the EPE (a Monte Carlo estimate with $n = 4000$ total points) for different regions $R$ with $\lambda(t, x) = \exp(10x - 2 \cdot \mathbb{1}\{x \geq 0.4\}$ with a one-dimensional feature $x \in [0, 1]$. The region $[a, b]$ corresponds to the box whose bottom-left corner is at coordinate $(a, b)$ in the graph with the heat map value determined by the EPE for this region. Regions below the horizontal dashed red line are subsets of the left subinterval $[0, 0.4]$ (and therefore a Cox model holds in these regions); regions to the right of the vertical dashed red line are subsets of the right subinterval $[0.4, 1]$ (and therefore a Cox model holds in these regions as well, though not with the same baseline hazard function as the left subinterval). Regions in the upper-left quadrant contain points in both the left and right subintervals, and therefore the Cox model does not hold. The largest region in which the Cox model is well-specified and with the lowest EPE is $[0.4, 1]$, indicated by the cyan square. However, the minimum EPE is actually obtained by the whole interval $[0, 1]$, indicated by the red square.

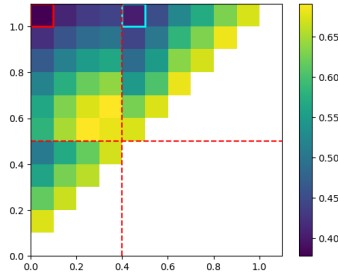

*Figure 1.* Counterexample motivating the CRS. We use a Monte Carlo estimate for the EPE with $n = 4000$ total points. The value of the cell with bottom-left corner at coordinate $(a, b)$ corresponds to the EPE of the region $[a, b]$, and the color map shows the value of the EPE. The interval $[c, 1]$ is the region of minimal EPE in which the Cox model holds (cyan boxed cell). However, the region which minimizes the EPE overall is the entire interval $[0, 1]$ (red boxed cell), which does not follow a Cox model.

In spite of this example, the EPE is still useful for two reasons. First, although the Cox model is not well-specified on the minimizing region, it will still make good predictions. Second, while the CRS is able to detect deviations from the Cox model, it is not immediately clear how to turn the CRS into a single value for a group of points.

## 4.2. CRS Definition

We first consider uncensored data. Let $\beta$ be the fitted model coefficients and $x_1, \ldots, x_n$ be the feature vectors in the core group, labeled such that $t_1 < t_2 < \cdots < t_n$. For a "test" point with features $x^*$ and failure time $t^*$, we compute the probability that the rank of $x^*$ is at least as extreme (high or low) as its observed value, conditional on the other observed failure times and assuming that $x^*$ follows the same Cox model as the core group. To do this, we define the *conditional rank statistics* of $x^*$, given by $r_k^c(x^*) =$

$$\mathbb{P}(t_{k-1} < t^* < t_k \mid x^*, x_1, \ldots, x_n; \, t_1 < \cdots < t_n), \quad (5)$$

where the probability is computed assuming each pair $(x, t)$ follows the same Cox model with fixed (unknown) baseline hazard function $\lambda_0(t)$ and Cox coefficients $\beta$. We also define the *unconditional rank probabilities* of $x^*$ as $r_k(x^*) =$

$$\mathbb{P}(t_1 < \cdots < t_{k-1} < t^* < t_k < \cdots < t_n | x^*, x_1, \ldots, x_n). \quad (6)$$

By Bayes' rule, $r_k^c(x^*) = r_k(x^*)/(\sum_{j=1}^{n+1} r_j(x^*))$, so it suffices to compute the $r_k(x^*)$. When the data are generated according to the Cox model,

$$r_k(x^*) = \prod_{i=1}^{n+1} \frac{\exp(\beta^\top x_i^{(k)})}{\sum_{j=i}^{n+1} \exp(\beta^\top x_j^{(k)})}, \quad (7)$$

where we have defined $x_i^{(k)} = x_i$ if $i < k$, $x_k^{(k)} = x^*$, and $x_i^{(k)} = x_{i-1}$ if $i > k$.

**Generalization to Censored Data** The conditional rank statistics have a straightforward generalization to the partial likelihood and censored data. Let $t_1 < \cdots < t_n$ be the event times for the points with features $x_1, \ldots, x_n$ in the core group, and let $\delta_i$ be the corresponding failure indicators ($\delta_i = \mathbb{1}\{x_i$ failed (was not censored) at time $t_i\}$). The partial likelihood that $x^*$ fails with event rank $k$ is

$$r_k(x^*) = \prod_{i \, : \, \delta_i^{(k)} = 1} \left( \frac{\exp(\beta^\top x_i^{(k)})}{\sum_{j=i}^{n+1} \exp(\beta^\top x_j^{(k)})} \right), \quad (8)$$

where $x_i^{(k)}$ are defined as before and $\delta_i^{(k)}$ are defined analogously: $\delta_i^{(k)} = \delta_i$ for $i < k$, $\delta_k^{(k)} = 1$, and $\delta_i^{(k)} = \delta_{i-1}$ for $i > k$. Note that this is simply the standard Cox partial likelihood if $x^*$ fails as the $k$-th event. The conditional failure "likelihoods" $r_k^c(x^*)$ are then defined analogously to the case with no censoring, i.e., $r_k^c = r_k / \sum_{j=1}^{n+1} r_j$. We note that these are no longer actually probabilities in the presence of censoring.

**Fast Implementation** Computing the conditional rank probabilities naively is inefficient on large datasets, scaling as $\Omega(n^3)$. Using some recursive relationships between the unconditional rank probabilities, we can drastically reduce this runtime down to $O(n)$ which also leads to marked practical efficiency gains. Details can be found in Appendix C.

### 4.3. Properties of the CRS

We show that the CRS admit a "conditional Glivenko-Cantelli theorem," meaning that in the absence of censoring, the CRS converges uniformly to its large-sample expectation in probability. This result allows us to analyze the effect size which the CRS can detect given a large sample. The proof can be found in Appendix E.

**Theorem 4.1** (Informal version of Theorem E.1). *Fix a region $R$ of the feature space and assume there is no censoring in the data. Let $P_R$ denote the joint distribution of $(X, T(X))$ conditional on $X \in R$. Let $G(t) = \mathbb{P}(T(X) \leq t \mid X \in R)$ be the cdf of the marginal survival time distribution for $P_R$. Let $\widetilde{G}_n(\alpha) = \sum_{k=1}^{\lfloor \alpha n \rfloor} r_k^c(x^*)$ be the "cdf" of the CRS with respect to $n$ i.i.d. samples from $P_R$. Let $x^*$ be a fixed test point. As $n \to \infty$, uniformly in $\alpha$ we have $\widetilde{G}_n(\alpha) \xrightarrow{p} \mathbb{P}(G(T(x^*)) \leq \alpha)$.*

## 5. Algorithms

We introduce eight algorithms in total (not including the base method of fitting the Cox model to the entire dataset). Given a training dataset and hyperparameter settings, each method returns a region $R$ which defines the subgroup, after which the Cox coefficients $\beta$ are fit to the training data in this subgroup. Each method has a total of 100 possible hyperparameter settings which are discussed in Appendix H.1 along with implementation details for the algorithms.

**Base:** The simplest baseline is the standard Cox model fit to the entire dataset. The "subgroup" $R$ in this case is just the bounding box for the entire dataset.

**Random:** This method randomly selects $2d$ points (where $d$ is the data dimension) and forms their bounding box to determine $R$. The number $2d$ is chosen so that each randomly selected point determines one side of the bounding box.

**Survival Tree (ST):** Survival trees (Ishwaran et al., 2008) are a classical tree-based method for survival analysis where the node splitting is performed according to the log-rank criterion. We use the tree's leaves to define candidate subgroups and select $R$ as the leaf with minimum training EPE.

**Optimal Sparse Survival Tree (OSST):** We also tested another tree-based method (Zhang et al., 2024). Compared to the regular survival tree, OSST produces sparse trees and replaces the greedy node-splitting rule with an algorithm which provably finds the optimal collection of splits. As with the standard ST, we select the leaf with minimum training EPE as the subgroup.

**Cox Tree (CT):** We grow a tree using the EPE as a splitting criterion. The impurity of a split is defined as the weighted average of the EPE in the left and right children (weighted according to the size of the children). $R$ is again chosen as

the leaf with minimum EPE.

**PRIM:** Introduced by Friedman & Fisher (1999), PRIM is a general-purpose subgroup discovery method for finding axis-aligned regions of the data where a pre-specified quality function is maximized. We adapt this method to the survival analysis setting by using the negative EPE as the quality.

**DDGroup (DG):** The DDGroup framework of (Izzo et al., 2023) can incorporate both the EPE and the CRS, using the EPE for the core group selection procedure and the CRS for the rejection/expansion phase. The algorithm has three phases: first, a core group of points is selected and used to obtain a rough model fit. Second, points which could not feasibly follow the same model as the core group are rejected. Finally, the subgroup is defined as a large region which contains no rejected points. In our setting, the core group is selected as a neighborhood of $n$ points with minimum EPE. For the rejection phase, given an uncensored test point, let $k^*$ be the rank of the test point's failure time among core group failure times and define the score $\tau^* = \min \left\{ \sum_{k=1}^{k^*} r_k^c(x^*), \sum_{k=k^*}^{n+1} r_k^c(x^*) \right\}$. For censored test points, we only have a lower bound on its rank. Thus we only use the right tail of the CRS for scoring: if $k^*$ is the rank of the censoring time for the test point with features $x^*$, we set $\tau^* = \sum_{k=k^*}^{n+1} r^c(x^*)$. In either case, the rejection label is defined as $\mathbb{1}\{\tau^* < \alpha\}$ for a rejection threshold $\alpha$.

We also test two ablations of DDGroup which make use of the C-index (**DG-CI**) or partial likelihood (**DG-PL**) to test the value of the EPE and CRS from an algorithmic perspective. Details on the ablations can be found in Appendix H.1.7 and in the supplementary code. Pseudocode for all of the methods is provided in Appendix H.1.

## 6. Performance Guarantees

It is possible to show that DDGroup, recovers the correct region in a well-specified setting. This result relies on several assumptions: (1) The hazard function for the entire dataset has the form $\lambda(t; x) = \lambda_0(t) e^{h(x)}$ for some unknown risk function $h$. (2) There is no censoring in the data. (3) There is a unique largest region $R^*$ which minimizes the EPE, and $R^*$ is an axis-aligned box. (4) Conditional on $x \in R^*$, we have $h(x) = \beta^\top x$ for some $\beta$, i.e., the Cox model is well-specified. (5) The core group selection procedure finds a group of points which belong to $R^*$, and the Cox model fit to these points recovers the true parameters $\beta$. (6) The error between the finite conditional rank statistics and its large-sample limit according to Theorem E.1 is negligible. Under these assumptions, we can analyze a "theoretically stylized" version of DDGroup, given by Algorithms 3 and 4 in the appendix. The proof, as well as a discussion on the validity of the assumptions, can be found in Appendix F.

**Theorem 6.1.** *Let $\hat{R}_n$ be the region output by Algorithm 4*

*on a dataset of $2n$ i.i.d. points satisfying Assumptions A1-A5. For any $\varepsilon > 0$, there is an effect size $C_\varepsilon = O(\log \varepsilon^{-1})$ such that if $|h(x) - \beta^\top x| \geq C_\varepsilon$ outside of $R^*$, then there exist settings for the hyperparameters of Alg. 4 such that with probability at least 0.99, $R^* \subseteq \hat{R}_n$ and $\mathrm{vol}(\hat{R}_n \setminus R^*) \leq C'\varepsilon$ for another constant $C'$ as $n \to \infty$.*

# 7. Experiments

We now test our methods empirically. Unless noted otherwise, we use the following setup. First, each method was supplied with a range of 100 total hyperparameter settings (specified in Section H.1). Given a hyperparameter setting and a training dataset, each method returns a (subgroup, Cox coefficients) pair. Thus, for each random train/test split of a given dataset, each method returns a list of at most 100 subgroups and associated Cox coefficients. (There may be fewer than 100 subgroups since some hyperparameter settings may fail.) From among these 100 subgroups, we filtered to those which contained at least 10% of the training data to prevent overfitting. The remaining subgroup with the lowest training EPE was selected for each method and all metrics are computed on a held-out test set. The results were averaged over 10 random train/test splits of the data.

## 7.1. Synthetic Data

For the synthetic data, we follow (Izzo et al., 2023) and define the precision and recall of an estimated region $R$ by Precision $= \frac{\mathrm{vol}(R \cap R^*)}{\mathrm{vol}(R)}$ and Recall $= \frac{\mathrm{vol}(R \cap R^*)}{\mathrm{vol}(R^*)}$. The F1 score is then defined in the usual manner as the harmonic mean of precision and recall. An F1 score closer to 1 means better recover of the ground truth, with perfect recovery iff F1 = 1. We perform experiments on two synthetic datasets: Synth-Counter and Synth-Nonlinear. Complete descriptions of these datasets can be found in Appendix H.3.

**Synth-Counter** The data in Synth-Counter is generated according to the example in Section 4.1. The purpose of this experiment is to show that the CRS can cope with this counterexample, while the C-index and partial likelihood cannot. For this dataset only, rather than selecting the region with minimum training EPE for each method (which would generally return the whole interval $\mathcal{B}$), we simply select the region with the highest F1 score among the 100 regions returned by the method for each of the 10 replicates.

The results are shown in Table 1. By leveraging the CRS, DDGroup can be tuned to recover the ground truth region $R^*$. Relying only on the C-index or partial likelihood, it is not possible to tune DDGroup to recover $R^*$.

**Synth-Nonlinear** With this dataset, we study the more common scenario where we will select a subgroup according to the minimum training EPE. The features are uniformly generated from $\mathcal{B} = [-1, 1]^d$. We set $R^* = [-1/6^{\frac{1}{d}}, 1/6^{\frac{1}{d}}]^d$,

*Table 1.* Best F1 score obtained by different implementations of DDGroup in the example from Fig. 1 (Synth-Counter). Numbers in parentheses are standard error. By using the CRS, it is possible to tune DDGroup to find the correct subgroup. Even with the best possible hyperparameter tuning, the other versions which do not take advantage of the EPE and CRS cannot recover $R^*$.

| Metric | Base | DG-PL | DG-CI | DG |
|---|---|---|---|---|
| F1 ($\uparrow$) | 0.75 (0.00) | 0.81 (0.10) | 0.76 (0.04) | 0.94 (0.01) |

*Table 2.* Comparison of methods on Synth-Nonlinear. Parentheses show standard error. DDGroup (DG) obtains the best performance in terms of recovering $R^*$. The other methods have worse performance, though all methods except ST and PRIM outperform the non-subgroup baseline.

| Method | F1 ($\uparrow$) | EPE ($\downarrow$) | C-index ($\uparrow$) |
|---|---|---|---|
| Base | 0.29 (0.00) | 0.69 (0.00) | 0.54 (0.00) |
| ST | 0.28 (0.03) | 0.66 (0.01) | 0.59 (0.01) |
| OSST | 0.29 (0.01) | 0.69 (0.00) | 0.54 (0.00) |
| PRIM | 0.30 (0.01) | 0.69 (0.00) | 0.54 (0.00) |
| CT | 0.33 (0.08) | 0.64 (0.04) | 0.60 (0.03) |
| DG-PL | 0.46 (0.03) | 0.64 (0.02) | 0.61 (0.02) |
| Random | 0.75 (0.04) | 0.56 (0.03) | 0.74 (0.02) |
| DG-CI | 0.89 (0.03) | 0.39 (0.02) | 0.86 (0.01) |
| **DDGroup** | 0.97 (0.01) | 0.38 (0.02) | 0.87 (0.01) |

and conditional on $X \in R^*$ the survival time $T$ is generated according to a Cox model with baseline hazard $\lambda_0(t) \equiv 1$ and some ground truth Cox coefficients $\beta^*$. For $X \notin R^*$, $T|X$ also follows a Cox model but with respect to a nonlinear transformation of the features $X$.

The results are shown in Table 2. DDGroup has the best performance across all metrics, obtaining the best recovery of $R^*$ as well as the best predictive performance. All methods except ST, OSST, and PRIM significantly improve over the non-subgroup approach in all metrics. We ran an additional experiment using this setup in dimension $d = 4$; these results can be found in Table 5.

## 7.2. Real Data

In this subsection, we apply the methods to real data.

**Additional Metrics** As discussed in Section 4, practitioners may be interested in not only high predictability of the data (represented by low EPE) but also a qualitatively good fit of the Cox model to the data. When a ground truth region in which a Cox model is well-specified is unknown, we must resort to an observable proxy. Thus, we define the *rejection fraction*, which measures the number of individual points in a subgroup which deviate from the Cox model based on the CRS. See Appendix H for the definition.

**Datasets** We used datasets from the sksurv Python package (Pölsterl, 2020). Each of these datasets studies patient survival in clinical settings. A common design choice in clinical statistical analyses is to adjust for a single covariate

to observe its effect, and generally simple subgroup definitions are preferred (Friedman et al., 2015). Thus, in each of these datasets, we adjust for a single covariate and define a subgroup in terms of age. An explanation of how each algorithm is modified to account for different variables used to define the subgroups vs. fit the Cox model is given in Appendix H.1. We chose age as the subgroup-defining variable as age is known to modulate/interact with other biomarkers for many health-related outcomes (Belloy et al., 2023; Mak et al., 2023; Moqri et al., 2023). In Table 3, the results are presented as (base dataset name)-(adjusted covariate). Dataset descriptions are provided in Appendix H.3.

**Results** DDGroup finds subgroups with the lowest EPE in 2/3 datasets and also tends to have a low rejection fraction, indicating a small fraction of "outliers" within the subgroup. In fact, DDGroup is on the Pareto frontier of (EPE, rejection fraction) for every dataset. We note that the EPE of a null Cox model (i.e., $\beta = 0$, giving random guesses) is $-\log \frac{1}{2} \approx 0.69$; in some datasets, some baselines perform no better than random. Results on 3 additional datasets can be found in Appendix H.5. DDGroup remains on the Pareto frontier for these datasets.

*Table 3.* Results on the real datasets. The best value for each metric is highlighted in green. DDGroup and its variants tend to find the subgroup with lowest EPE and offers a balance between EPE and Cox model fit. The other baselines are often capable of improving over the basic Cox model fit to the entire data, though in some cases they do not perform better than a null model.

| | | Base | Rand | PRIM | ST | OSST | CT | DG-PL | DG-CI | DG |
|---|---|---|---|---|---|---|---|---|---|---|
| AIDS-Karnof | EPE (↓) | 0.62 (0.03) | 0.58 (0.11) | 0.62 (0.03) | 0.65 (0.07) | 0.62 (0.03) | 0.72 (0.22) | 0.62 (0.04) | 0.72 (0.22) | 0.38 (0.07) |
| | Rej@10% (↓) | 0.09 (0.01) | 0.01 (0.01) | 0.15 (0.01) | 0.11 (0.03) | 0.15 (0.01) | 0.19 (0.02) | 0.07 (0.01) | 0.06 (0.01) | 0.01 (0.01) |
| | C-Index (↑) | 0.66 (0.03) | 0.68 (0.07) | 0.66 (0.03) | 0.64 (0.05) | 0.66 (0.03) | 0.69 (0.07) | 0.66 (0.03) | 0.69 (0.07) | 0.84 (0.05) |
| | Size (↑) | 1.00 (0.00) | 0.13 (0.01) | 0.96 (0.02) | 0.21 (0.09) | 1.00 (0.00) | 0.44 (0.10) | 0.82 (0.12) | 0.65 (0.14) | 0.15 (0.01) |
| GBSG2-tsize | EPE (↓) | 0.68 (0.00) | 0.62 (0.04) | 0.68 (0.00) | 0.68 (0.02) | 0.68 (0.00) | 0.63 (0.04) | 0.68 (0.00) | 0.65 (0.03) | 0.61 (0.04) |
| | Rej@10% (↓) | 0.14 (0.00) | 0.13 (0.03) | 0.08 (0.01) | 0.04 (0.01) | 0.08 (0.01) | 0.05 (0.02) | 0.13 (0.02) | 0.11 (0.02) | 0.07 (0.02) |
| | C-Index (↑) | 0.57 (0.01) | 0.62 (0.04) | 0.55 (0.01) | 0.57 (0.03) | 0.57 (0.01) | 0.62 (0.04) | 0.58 (0.01) | 0.64 (0.05) | 0.64 (0.04) |
| | Size (↑) | 1.00 (0.00) | 0.15 (0.02) | 0.94 (0.01) | 0.26 (0.07) | 1.00 (0.00) | 0.11 (0.01) | 0.90 (0.10) | 0.39 (0.14) | 0.12 (0.01) |
| VLC-Karnof | EPE (↓) | 0.57 (0.02) | 0.42 (0.14) | 0.58 (0.02) | 0.45 (0.10) | 0.57 (0.02) | 0.22 (0.05) | 0.32 (0.02) | 0.34 (0.06) | 0.33 (0.06) |
| | Rej@10% (↓) | 0.04 (0.01) | 0.12 (0.04) | 0.03 (0.01) | 0.07 (0.03) | 0.04 (0.01) | 0.21 (0.04) | 0.11 (0.05) | 0.10 (0.03) | 0.09 (0.03) |
| | C-Index (↑) | 0.69 (0.02) | 0.84 (0.06) | 0.68 (0.02) | 0.77 (0.07) | 0.70 (0.02) | 0.93 (0.02) | 0.92 (0.03) | 0.87 (0.03) | 0.87 (0.04) |
| | Size (↑) | 1.00 (0.00) | 0.17 (0.02) | 0.88 (0.05) | 0.23 (0.05) | 1.00 (0.00) | 0.20 (0.02) | 0.23 (0.03) | 0.22 (0.02) | 0.28 (0.08) |

*Table 4.* Results on the NASA dataset. The best EPE, C-index, and precision values are highlighted in green. The highest recall among those methods with perfect precision (Cox tree) is also highlighted in green. Several methods are able to discover subgroups which agree with the known ground truth operating conditions and which reveal qualitatively different behavior from the overall dataset.

| | Base | Rand | PRIM | ST | OSST | CT | DG-PL | DG-CI | DG |
|---|---|---|---|---|---|---|---|---|---|
| EPE (↓) | 0.65 (0.00) | 0.53 (0.01) | 0.65 (0.00) | 0.53 (0.01) | 0.65 (0.01) | 0.52 (0.01) | 0.64 (0.01) | 0.54 (0.01) | 0.53 (0.01) |
| C-Index (↑) | 0.60 (0.00) | 0.74 (0.01) | 0.60 (0.00) | 0.73 (0.01) | 0.60 (0.00) | 0.74 (0.01) | 0.62 (0.01) | 0.73 (0.01) | 0.73 (0.01) |
| Precision (↑) | 0.63 (0.00) | 1.00 (0.00) | 0.64 (0.01) | 1.00 (0.00) | 0.63 (0.01) | 1.00 (0.00) | 0.67 (0.04) | 1.00 (0.00) | 1.00 (0.00) |
| Recall (↑) | 1.00 (0.00) | 0.24 (0.02) | 0.91 (0.06) | 0.40 (0.06) | 1.00 (0.00) | 0.73 (0.10) | 0.92 (0.08) | 0.29 (0.04) | 0.36 (0.08) |
| $\beta_{Nc}$ | 0.31 (0.01) | 2.35 (0.10) | 0.32 (0.01) | 2.00 (0.09) | 0.31 (0.01) | 2.19 (0.08) | 0.53 (0.22) | 2.17 (0.09) | 2.23 (0.09) |
| $\beta_{NRf}$ | 0.30 (0.01) | -1.19 (0.07) | 0.30 (0.01) | -0.94 (0.10) | 0.30 (0.01) | -1.08 (0.05) | 0.15 (0.15) | -1.06 (0.06) | -1.10 (0.06) |

**Case Study: NASA Jet Engine Data** To showcase the utility of the discovered subgroups, we conducted an experiment on the NASA C-MAPSS dataset (Saxena et al., 2008). The goal is to model how different sensor readings (engine core speed "Nc" and corrected fan speed "NRf") affect the remaining useful life (RUL) of jet engines. In this dataset, there are known operating conditions related to different altitudes where the relationship between sensor readings and RUL will differ qualitatively. These different conditions serve as natural ground truth subgroups of the data.

Refer to Table 4. The Cox tree performs best overall, achieving the lowest EPE and highest recall among methods with perfect precision (i.e., those which identify subgroups that correspond entirely to one ground-truth operating condition). The discovered high-altitude subgroups reveal a qualitative change in Cox coefficients: while the baseline model shows small positive coefficients for both speeds (reflecting

mechanical wear), the high-altitude subgroup has a larger positive coefficient for core speed but a negative coefficient for fan speed, suggesting that at high altitude, increased fan usage may reduce engine strain compared to core reliance. This suggests an actionable plan for engine usage at high altitudes: increase reliance on the fan, and reduce reliance on the core. This is the trend in modern jet engines which have a reduced core size (see, e.g., this NASA article). A complete discussion of the dataset and results can be found in Appendix H.6.

## 8. Conclusion

We introduced the problem of subgroup discovery with the Cox model, where the goal is to find interpretable subsets of the feature space in which a Cox model makes confident and accurate predictions, and proposed several methods for this problem. The most successful method relies on two

components: the expected prediction entropy (EPE), which quantifies the ability of a survival model to discriminate between the relative risk of failure for two units; and the conditional rank statistics (CRS), a statistical object which can be used to measure the deviation of an individual datapoint to the distribution of survival times in an existing subgroup. We studied the theoretical properties of these methods and metrics and confirmed their effectiveness empirically on synthetic and real datasets.

**Limitations & Future Work** There is an intricate dependence between the feature distribution and the value of the EPE. Devising a way to disentangle the effect of the feature distribution from the intrinsic performance of the model itself on the EPE would provide better "apples-to-apples" comparisons between the value of different subgroups. Another important open direction is to create valid p-values for Cox models and subgroups discovered with our methods. We have partially alleviated this issue by using a held-out test set, but more advanced methods tailored to the subgroup problem may give more precise control of false discoveries.

## Impact Statement

Our methods can be considered as data-driven hypothesis generation methods, and as such are susceptible to false positives, i.e., "discovering" a subgroup which does not really exist. If acted upon without independent validation, this could lead to patients being inappropriately included in or excluded from treatments in a clinical setting. We therefore emphasize that discovered subgroups should be treated as hypotheses for further investigation, not as definitive guidance.

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

## A. Background on Survival Analysis and the Cox Model

A *survival time* is a non-negative random variable $T$ which describes the amount of time until an event of interest. Examples of commonly modeled events include the onset of a disease, the death of a patient, the time at which a customer stops using a product or platform, or the failure of a mechanical component. The arbitrary event to be modeled is referred to as a *failure*. Unlike more typical regression tasks in machine learning where the goal is to give a point estimate of a continuous-valued target, the goal of survival analysis is usually to model the *distribution* of $T$ conditional on some associated covariates $X \in \mathbb{R}^d$.

Natural modeling targets for describing the distribution of $T$ include standard probabilistic quantities such as the probability density function (pdf) or cumulative distribution function (cdf) of $T$, conditional on the features $X$, and indeed some survival analysis methods take this approach. A more common target, however, is the *hazard function*, defined as

$$\lambda(t;x) = \lim_{dt \to 0} \frac{\mathbb{P}(t \leq T \leq t + dt \mid T \geq t, X = x)}{dt}. \tag{9}$$

The hazard function can be thought of as an instantaneous rate of failure in the infinitesimal time interval $[t, t + dt)$, conditional on surviving up to time $t$ and on the features $X = x$. The hazard function is related to more standard quantities like the pdf or cdf. Specifically, letting $F(t, x) = \mathbb{P}(T \leq t \mid X = x)$ be the cdf and $f(t, x)$ the associated pdf (assuming one exists), we have the following identities:

$$S(t, x) := 1 - F(t, x) = \exp\left\{ -\int_0^t \lambda(u, x)\, du \right\},$$

$$f(t, x) = \lambda(t, x) S(t, x).$$

The complement $S(t, x)$ of the cdf is referred to as the *survival function*. The existence of these formulas shows that determining the hazard function completely specifies the distribution of $T|X$, as it completely specifies the pdf or cdf. In a biomedical context, the hazard function has several advantageous properties which make it a natural modeling target, including but not limited to interpretability. For instance, a patient in remission from cancer would naturally be more interested in knowing the conditional probability of a recurrence given that they have not experienced one yet, rather than an absolute probability which is more easily described by the cdf (Tian, 2015).

The Cox model posits a particular semiparametric form for the hazard function which implies that a unit change in each covariate has a multiplicative effect on the hazard function, i.e.,

$$\lambda(t;x) = \lambda_0(t) \exp(\beta^\top x) \tag{10}$$

for some coefficients $\beta$. Note that the higher the value of the log relative hazard $\beta^\top x$, the faster the model predicts the unit with features $x$ will fail. In particular, this means that if $\beta^\top x > \beta^\top x'$, then the model predicts that the survival time for $x$ will be less than that of $x'$ ($x$ will fail first).

Let $L_i$ denote the index of the individual who fails at time $t_i$. Let $T_\ell$ denote the random failure time for the $\ell$-th individual, and define $R(t)$ to be the *risk set* at time $t$, i.e. the set of individuals $R(t) = \{\ell : T_\ell \geq t\}$ who have not failed before time $t$. It can be shown that

$$\mathbb{P}(L_i = \ell \mid T_{L_i} = t_i, R(t_i) = R_i) = \frac{\exp(x_\ell^\top \beta)}{\sum_{j \in R_i} \exp(x_j^\top \beta)}. \tag{11}$$

Given the failure times $t_1 \leq \ldots \leq t_n$ and associated features $x_i$ and failure indicators $\delta_i$, (Cox, 1972) then proposed estimating $\beta$ by maximizing the log partial likelihood

$$\mathcal{L}(\beta) := \sum_{1 \leq i \leq n,\, \delta_i = 1} \left[ x_i^\top \beta - \log\left( \sum_{j \geq i} \exp(x_j^\top \beta) \right) \right]. \tag{12}$$

While each term in the partial likelihood is a likelihood in the traditional sense, (Cox, 1975) showed that $\exp(\mathcal{L}(\beta))$ is *not* a marginal or conditional likelihood (unless one makes restrictive assumptions on the censoring patterns/failure times). Nevertheless, maximizing (12) still enjoys many of the same properties as traditional MLE, such as asymptotic normality and consistency (Cox, 1975).

# B. Expanded Related Work

The Cox model (Cox, 1972; 1975) is a standard method for survival analysis, and it has found widespread use in practice due to its ease of interpretation. Nevertheless, the interpretability comes at the cost of strong modeling assumptions which may be violated in practice (Hernán, 2010). Within the machine learning community, there has been a great deal of effort to apply modern ML techniques to survival data and provide more powerful and flexible models (Katzman et al., 2018; Hu & Nan, 2023; Wu et al., 2023; Bleistein et al., 2024; Bertsimas et al., 2022; Kim, 2023; Chen, 2024; Lee et al., 2018; Rindt et al., 2022; Vauvelle et al., 2023; Wang et al., 2019).

One of the most common evaluation metrics for survival models is Harrell's concordance index (C-index) (Harrell et al., 1982), which evaluates a model according to how well the predicted failure order of the units matches the data. There are many known shortcomings of relying solely on the C-index for survival model evaluation (Hartman et al., 2023), and the examples raised in this paper add to the body of evidence that these are valid concerns. Thus, in addition to improving modeling flexibility, there has been recent interest in proper evaluation metrics for survival models (Yanagisawa, 2023; Haider et al., 2020; Qi et al., 2023).

Our work sits at the intersection of two orthogonal topics, survival analysis and subgroup discovery. At a high level, subgroup discovery refers to mining datasets for subsets or regions in which the data distribution is in some sense "interesting," usually quantified by a numerical score function taking an extreme value when evaluated on the subgroup (Friedman & Fisher, 1999; Atzmueller, 2015; Leman et al., 2008; Izzo et al., 2023; Xu et al., 2024). While subgroup discovery is a general problem, it has found a great deal of applications in biostatistics (Lipkovich et al., 2017a; 2023). Many methods have been proposed to study heterogeneous treatment effects in patient populations, in particular to find patient groups which experience enhanced benefit from a treatment (Kehl & Ulm, 2006; Lipkovich et al., 2011; Dusseldorp & Van Mechelen, 2014; Lipkovich & Dmitrienko, 2014; Lipkovich et al., 2017b; Schnell et al., 2018; Schnell, 2021; Li & Imai, 2025); or for purposes of patient stratification (Polonik & Wang, 2010; Chen et al., 2015; Huang et al., 2017).

The work most closely related to ours in spirit is (Wei & Kosorok, 2018), which also studied a subgroup discovery problem for the Cox model. The resulting model, called the change-plane Cox model, defines subgroups via the two sides of a hyperplane rather than an axis-aligned box. There are also more restrictive assumptions on the relationship between the two resulting subgroups in their setting than in the setting considered in the present paper. These differences are quite significant and render their method unsuitable for our setting; we give an in-depth discussion in Appendix H.1. (Zhang et al., 2025) also studied properties of this change-plane Cox model.

# C. Runtime Improvements

A naive implementation of the conditional rank tail probability took over 20 seconds to evaluate on a single point in some early experiments. Thus, a faster implementation is necessary. We will use the abbreviation $r_k = r_k(x^*)$.

First, we observe that the naive computation of a single $r_k$ from equation (7) will require $\Omega(n^2)$ time. This can easily be reduced to $O(n)$ by updating the partial sum contained in the denominator as each term in the product is computed, rather than recomputing it from scratch each time. With this modification, we can compute $r_1$ in $O(n)$ time.

We can obtain another speedup by computing the remaining $r_k$ recursively, rather than repeatedly using the procedure above from scratch for each $r_k$. A direct calculation using the formula (8) shows that

$$r_{k+1} = \frac{(1 - \delta_k)e^{\beta^\top x^*} + S_k}{e^{\beta^\top x^*} - e^{\beta^\top x_k} + S_k} \cdot r_k, \tag{13}$$

where we have defined $S_k = \sum_{i=k}^{n} e^{\beta^\top x_i}$. Again using the running partial sum trick to quickly compute $S_k$ (rather than computing from scratch each time), we can compute the next $r_{k+1}$ in constant time using the previous one. This means that $r_1, \ldots, r_{n+1}$ can *all* be computed using only $O(n)$ time total.

The pseudocode for the resulting procedure is given in Algorithm 1. We have replaced the rank probabilities $r_k$ with the logarithms since when working with large datasets, working directly with the product of many probabilities (even when each is individually of "reasonable" size) can lead to numerical issues. Given the set of $\log r_k$, the CRS $r^c$ can then be computed by taking a softmax.

---

**Algorithm 1** Fast computation of the log rank probabilities with censoring

---

$S \leftarrow \sum_{i=1}^{n} e^{\beta^\top x_i}$

$\text{log\_prod} \leftarrow \beta^\top x^* - \log(S + e^{\beta^\top x^*})$

**for** $i = 1, \ldots, n$ **do**

    $\text{log\_prod} \leftarrow \text{log\_prod} + \delta_i(\beta^\top x_i - \log S)$

    $S \leftarrow S - e^{\beta^\top x_i}$

**end for**

$\log r_1 \leftarrow \text{log\_prod}$

$S \leftarrow \sum_{i=1}^{n} e^{\beta^\top x_i}$

**for** $k = 1, \ldots, n$ **do**

    $\log r_{k+1} \leftarrow \log r_k + \log(S + (1 - \delta_k)e^{\beta^\top x^*}) - \log(S + e^{\beta^\top x^*} - e^{\beta^\top x_k})$

    $S \leftarrow S - e^{\beta^\top x_k}$

**end for**

    **return** $\log r_1, \ldots, \log r_{n+1}$

---

## D. Properties of the EPE

### D.1. EPE Is a Proper Scoring Rule

In this subsection, we give the proof of Proposition 3.2, restated here for convenience.

**Proposition 3.2.** *Suppose that the ground truth hazard function follows the Cox model, i.e., $\lambda(t, x) = \lambda_0(t)e^{\beta^\top x}$. Then the EPE is minimized iff $\hat{\beta}^\top(X - X') = \beta^\top(X - X')$ with probability 1.*

*Proof.* The cross-entropy $H(p, q) = -p \log q - (1 - p) \log(1 - q)$ is a strictly proper scoring rule, i.e., for $p$ fixed it is minimized iff $q = p$. By (1), after marginalizing over the distribution of $Y$, the EPE (3) can be written as

$$\text{EPE}(\hat{\beta}, R) = \mathbb{E}\left[ H(\sigma(\beta^\top(X - X')), \sigma(\hat{\beta}^\top(X - X'))) \,\Big|\, X, X' \in R \right],$$

where $\sigma(z) = 1/(1 + e^{-z})$ is the sigmoid function. By the fact that the cross-entropy is a strictly proper scoring rule, it follows that $\sigma(\beta^\top(X - X')) = \sigma(\hat{\beta}^\top(X - X'))$ almost surely. The result then follows since the sigmoid function is injective. $\square$

### D.2. EPE Decreases with Region Size

In this subsection, we give the proof of Theorem 3.3.

**Lemma D.1.** *Let $H(z) = -\frac{1}{1+e^{-z}} \log(\frac{1}{1+e^{-z}}) - \frac{1}{1+e^z} \log(\frac{1}{1+e^z})$. Then $H(z)$ is a decreasing function of $|z|$.*

*Proof.* By taking a derivative, we see that the function $h(p) := -p \log p - (1-p) \log(1-p)$ is increasing in $p$ for $p \in [0, 1/2]$ and decreasing in $p$ for $p \in [1/2, 1]$. Equivalently, $h(p)$ is a decreasing function of $|p - 1/2|$. Setting $p(z) = \frac{1}{1+e^{-z}}$, we see that $p(z) = 1/2$ for $z = 0$ and $p(z)$ moves away from $1/2$ (i.e., $|p(z) - 1/2|$ increases) as the magnitude of $z$ increases. This yields the desired result. $\square$

**Lemma D.2.** *Let $Z, Z'$ be random variables with densities $f, g$ respectively. Suppose that there exists a constant $c \geq 0$ such that $f(z) \geq g(z)$ whenever $|z| < c$ and $f(z) \leq g(z)$ whenever $|z| \geq c$. Then $\mathbb{E}[H(Z)] \geq \mathbb{E}[H(Z')]$.*

*Proof.* We have the following:

$$\mathbb{E}[H(Z)] - \mathbb{E}[H(Z')] = \int_{|z|<c} H(z)(f(z) - g(z))\, dz + \int_{|z|\geq c} H(z)(f(z) - g(z))\, dz$$

$$\geq \inf_{|z|<c} H(z) \cdot \int_{|z|<c} (f(z) - g(z))\, dz - \sup_{|z|\geq c} H(z) \cdot \int_{|z|\geq c} (g(z) - f(z))\, dz \tag{14}$$

$$= \inf_{|z|<c} H(z) \cdot (\mathbb{P}(|Z| < c) - \mathbb{P}(|Z'| < c)) - \sup_{|z|\geq c} H(z) \cdot (\mathbb{P}(|Z'| \geq c) - \mathbb{P}(|Z| \geq c))$$

$$= (\inf_{|z|<c} H(z) - \sup_{|z|\geq c} H(z))(\mathbb{P}(|Z| < c) - \mathbb{P}(|Z'| < c)) \tag{15}$$

$$\geq 0. \tag{16}$$

Equation (14) holds because $f(z) - g(z) \geq 0$ when $|z| < c$ and $f(z) - g(z) \leq 0$ when $|z| \geq c$. Note that this also implies that $\mathbb{P}(|Z| < c) - \mathbb{P}(|Z'| < c) = \int_{|z|<c}(f(z) - g(z))\, dz \geq 0$. Equation (15) holds by substituting $\mathbb{P}(|Z| \geq c) = 1 - \mathbb{P}(|Z| < c)$ and similarly for $Z'$. Equation (16) holds because $\inf_{|z|<c} H(z) \geq \sup_{|z|\geq c} H(z)$ by Lemma D.1, and because $\mathbb{P}(|Z| < c) - \mathbb{P}(|Z'| < c) \geq 0$ as established previously. This completes the proof. $\square$

**Proposition 3.3.** *Let the joint data distribution $P$ be such that the marginal distribution of the features $P|_X$ is uniform on a region $\mathcal{B} \subseteq \mathbb{R}^d$. Let $R = \prod_{i=1}^{d}[a_i, b_i]$, $R' = \prod_{i=1}^{d}[a'_i, b'_i]$ be axis-aligned boxes such that $R, R' \subseteq \mathcal{B}$ and such that $|a_i - b_i| \leq |a'_i - b'_i|$ for all $i$. Further suppose that $T|X$ follows a Cox model with coefficients $\beta$ whenever $X \in R \cup R'$. Then $\mathrm{EPE}(\beta, R') \leq \mathrm{EPE}(\beta, R)$.*

*Proof.* By marginalizing (3) over $Y$, we have that

$$\mathrm{EPE}(\beta, R) = \mathbb{E}[H(\beta^\top (X - X')) \mid X, X' \in R]$$

and similarly for $R'$. Note that since the expectation depends only on $X - X'$, it is translation invariant, so we may assume that $R = \prod_{i=1}^{d}[0, c_i]$ and $R' = \prod_{i=1}^{d}[0, c'_i]$. We may further assume that $c_i = c'_i$ for all $i \geq 2$ (i.e., $R$ and $R'$ differ in only a single side length): if we can show that the EPE decreases when only one side length is increased, then we can create a chain of at most $d$ inequalities in the EPE (where one side length increases at time) to prove the general inequality. Finally, since the expectation depends only on $\beta^\top (X - X')$, we can replace $\beta_1$ (the first coordinate of $\beta$) with $c_1 \beta_1$ and $X_1, X'_1$ (the first coordinates of $X$ and $X'$) with $X/c_1, X'/c_1$. This effectively replaces $c_1$ with $c_1/c_1 = 1$ and $c'_1$ with $c'_1/c_1 = c > 1$. Thus, we may assume that $c_1 = 1$ and $c'_1 = c > 0$.

Let $Z = \beta^\top (X - X')$ for $X, X' \sim \mathrm{Unif}(R)$ and $Z' = \beta^\top (X - X')$ for $X, X' \sim \mathrm{Unif}(R')$, and let $f, g$ be the densities for $Z, Z'$ respectively. By Lemma D.2, it suffices to show that there is a constant $c$ such that $f(z) \geq g(z)$ for $|z| < c$ and $f(z) \leq g(z)$ for $|z| \geq c$. Since the distributions of $Z$ and $-Z$ are equal (and similarly for $Z', -Z'$), $f$ and $g$ are both symmetric so it suffices to show this for $z \geq 0$.

For any $c > 0$, define $\psi_c(z) = \frac{c - |z|}{c^2}$. For a $d$-dimensional vector $v$, let $v_{\backslash 1}$ denote the $d - 1$-dimensional vector consisting of all but the first entry of $v$. Finally, define $\varphi(z)$ to be the density of $\beta_{\backslash 1}^\top (X_{\backslash 1} - X'_{\backslash 1})$ for $X_{\backslash 1}, X'_{\backslash 1} \sim \mathrm{Unif}(\prod_{i=2}^{d}[0, c_i])$. The densities of $Z$ and $Z'$ are given by convolutions: $f = \psi_{|\beta_1|c_1} * \varphi$ and $g = \psi_{|\beta_1|c'_1} * \varphi$. We will now proceed show that the ratio $g(z)/f(z)$ is nondecreasing for $z \geq 0$ by showing that

$$\frac{d}{dz}\left[\frac{g(z)}{f(z)}\right] \geq 0 \quad \text{for} \quad z \geq 0.$$

Since $g, f \geq 0$, this will suffice to prove the claim. (In particular, we can take the $c$ in Lemma D.2 to be $\sup\{z : g(z)/f(z) \leq 1\}$.) The derivative can be computed as follows:

$$\frac{d}{dz}\left[\frac{g}{f}\right] = \frac{d}{dz}\left[\frac{\psi_{|\beta_1|c'_1} * \varphi}{\psi_{|\beta_1|c_1} * \varphi}\right] = \frac{(\psi_{|\beta_1|c_1} * \varphi)(\psi'_{|\beta_1|c'_1} * \varphi) - (\psi_{|\beta_1|c'_1} * \varphi)(\psi'_{|\beta_1|c_1} * \varphi)}{(\psi_{|\beta_1|c_1} * \varphi)^2}. \tag{17}$$

Since the denominator of (17) is nonnegative, it suffices to show that the numerator is nonnegative. Note that $\frac{d}{dz}\psi_c(z) =$

$-\mathrm{sgn}(z)/c^2$. Thus, the numerator of (17) becomes

$$(\psi_{|\beta_1|c_1} * \varphi)(\psi'_{|\beta_1|c'_1} * \varphi) - (\psi_{|\beta_1|c'_1} * \varphi)(\psi'_{|\beta_1|c_1} * \varphi)$$

$$= (\psi_{|\beta_1|c_1} * \varphi)\left(\frac{-\mathrm{sgn}}{(|\beta_1|c'_1)^2} * \varphi\right) - (\psi_{|\beta_1|c'_1} * \varphi)\left(\frac{-\mathrm{sgn}}{(|\beta_1|c_1)^2} * \varphi\right)$$

$$= \left(\left(\frac{\psi_{|\beta_1|c_1}}{(|\beta_1|c'_1)^2} - \frac{\psi_{|\beta_1|c'_1}}{(|\beta_1|c_1)^2}\right) * \varphi\right)(-\mathrm{sgn} * \varphi). \tag{18}$$

A direct computation shows that

$$\frac{\psi_{|\beta_1|c_1}}{(|\beta_1|c'_1)^2} - \frac{\psi_{|\beta_1|c'_1}}{(|\beta_1|c_1)^2} = \frac{|\beta_1|c_1 - |z|}{(|\beta_1|c_1)^2(|\beta_1|c'_1)^2} - \frac{|\beta_1|c'_1 - |z|}{(|\beta_1|c'_1)^2(|\beta_1|c_1)^2} = \frac{|\beta_1|(c_1 - c'_1)}{(|\beta_1|c_1)^2(|\beta_1|c'_1)^2}.$$

Since $c'_1 \geq c_1$, this is a nonpositive constant function. Convolving against $\varphi$ (which is nonnegative function) will then result in a nonpositive constant.

Finally, we compute $-\mathrm{sgn} * \varphi$:

$$-(\mathrm{sgn} * \varphi)(z) = -\int \varphi(y)\mathrm{sgn}(z - y)\,dy$$

$$= \int_{y>z} \varphi(y)\,dy - \int_{y\leq z} \varphi(y)\,dy$$

$$= \mathbb{P}(\beta_{\backslash 1}^\top(X_{\backslash 1} - X'_{\backslash 1}) > z) - \mathbb{P}(\beta_{\backslash 1}^\top(X_{\backslash 1} - X'_{\backslash 1}) \leq z).$$

Since the distribution of $\beta_{\backslash 1}^\top(X_{\backslash 1} - X'_{\backslash 1})$ is symmetric about 0, it follows that

$$\mathbb{P}(\beta_{\backslash 1}^\top(X_{\backslash 1} - X'_{\backslash 1}) > z) \leq 1/2, \quad \mathbb{P}(\beta_{\backslash 1}^\top(X_{\backslash 1} - X'_{\backslash 1}) \leq z) \geq 1/2$$

for $z \geq 0$. Thus, $(-\mathrm{sgn} * \varphi)(z) \leq 0$ for $z \geq 0$.

To conclude, we have now shown that (18) is the product of two nonpositive quantities, therefore it must be nonnegative. This completes the proof. $\square$

## E. Properties of the Conditional Rank Statistics

In this section, we show that the CRS converge uniformly to their expectation in probability, which is the content of Theorem E.1.

**Theorem E.1.** *Fix a region $R$ of the feature space and assume there is no censoring in the data. Let*

$$G(t) = \mathbb{P}(T(X) \leq t \mid X \in R)$$

*be the cdf of the marginal core group survival time distribution, i.e., where we first sample random features $X$ from the core group, then sample $T|X$. Fix a test point $x^*$ and assume that $T^* := T(x^*)$ is absolutely continuous with bounded Radon-Nikodym derivative with respect to the marginal core group survival time distribution. Let*

$$\hat{G}_n(t) = \frac{1}{n}\sum_{i=1}^n \mathbb{1}\{t_i \leq t\}$$

*be the empirical cdf of the survival times given $n$ i.i.d. samples. In particular, in the case of no censoring,*

$$\mathbb{P}(\hat{G}_n(T^*) \leq \alpha | \{x_i\}\, t_1 < \cdots < t_n) = \sum_{k=1}^{\alpha n} r_k^c(x^*)$$

*is precisely the $\alpha$ tail of the CRS for the test point $x^*$. Then as $n \to \infty$, uniformly in $\alpha$ we have*

$$\mathbb{P}(\hat{G}_n(T^*) \leq \alpha | \{x_i\},\, t_1 < \cdots < t_n) \xrightarrow{p} \mathbb{P}(G(T(x^*)) \leq \alpha).$$

We remark that the absolute continuity and bounded density of $T^*$ with respect to the marginal core group survival distribution can be easily made to hold by imposing mild regularity conditions on the hazard function $\lambda(t, x)$ and the core group $R$.

*Proof.* By the Glivenko-Cantelli theorem, we have that $\sup_{t \in \mathbb{R}} |\hat{G}_n(t) - G(t)| \to 0$ almost surely, and therefore in probability as well.

Let $\mathcal{B}_x^n = \{\{x_i\}_{i=1}^n : \mathbb{P}_{\hat{G}_n}(\sup_{t \in \mathbb{R}} |\hat{G}_n(t) - G(t)| > \varepsilon \mid \{x_i\}_{i=1}^n) > \eta\}$. We claim that $\mathbb{P}(\{x_i\}_{i=1}^n \in \mathcal{B}_x^n) \to 0$ as $n \to \infty$ for any $\varepsilon, \eta > 0$. Let $\mathbb{P}(\mathcal{B}_x^n) = \gamma$. Again by Glivenko-Cantelli, for and $\varepsilon, \delta > 0$ we can choose $n$ large enough so that the following holds:

$$
\begin{aligned}
\delta &> \mathbb{P}(\sup_t |\hat{G}_n(t) - G(t)| > \varepsilon) \\
&= \mathbb{P}(\mathcal{B}_x^n)\mathbb{P}(\sup_t |\hat{G}_n(t) - G(t)| > \varepsilon \mid \mathcal{B}_x^n) + (1 - \mathbb{P}(\mathcal{B}_x^n))\mathbb{P}(\sup_t |\hat{G}_n(t) - G(t)| > \varepsilon \mid \neg\mathcal{B}_x^n) \\
&\geq \gamma\eta + (1 - \gamma) \cdot 0 = \gamma\eta.
\end{aligned}
\tag{19}
$$

If we take $\eta = \delta^{1/2}$ then we see that $\gamma \leq \delta^{1/2}$. Since $\delta$ can be made arbitrarily small, $\eta$ and $\gamma$ can also be made arbitrarily close to 0 for large enough $n$.

Choose $n$ large enough so that $\mathbb{P}(\{x_i\}_{i=1}^n \in \mathcal{B}_x^n) \leq \delta$, then consider a realization $\{x_i\}_{i=1}^n \notin \mathcal{B}_x^n$. By definition, we have

$$
\mathbb{P}(\sup_{t \in \mathbb{R}} |\hat{G}_n(t) - G(t)| > \varepsilon \mid \{x_i\}_{i=1}^n) \leq \delta.
\tag{20}
$$

Next, consider $N$ draws of the survival times conditional on the $x_i$, let $t_i^{(j)}$ be the survival time for $x_i$ in the $j$-th draw, and let $\hat{G}_n^{(j)}$ be the associated empirical cdf for the survival times. Let

$$
\mathcal{B}_d = \{j \in [N] : \sup_{t \in \mathbb{R}} |\hat{G}_n^{(j)}(t) - G(t)| > \varepsilon\}
$$

be the indices of the "bad" draws. By (20), $|\mathcal{B}_d| \leq \delta N + O(\sqrt{N})$ with high probability.

To simplify notation, let $T^* = T(x^*)$. We next claim the following: Given that $j \notin \mathcal{B}_d$, we have that $|\mathbb{P}_{T^*}(\hat{G}_n^{(j)}(T^*) \leq \alpha) - \mathbb{P}_{T^*}(G(T^*) \leq \alpha)| \leq 4C\varepsilon$ for all $\alpha$. To see this, observe that

$$
|\mathbb{P}_{T^*}(\hat{G}_n^{(j)}(T^*) \leq \alpha) - \mathbb{P}_{T^*}(G(T^*))| \leq \int_{t \geq 0} |\mathbb{1}\{\hat{G}_n^{(j)}(t) \leq \alpha\} - \mathbb{1}\{G(t) \leq \alpha\}| \, dP(t),
\tag{21}
$$

where $dP(t)$ denotes the probability measure for $T^*$. Since $j \notin \mathcal{B}_d$, we have $\sup_t |\hat{G}_n^{(j)}(t) - G(t)| \leq \varepsilon$. This implies the following:

- If $\hat{G}_n^{(j)}(t) \leq \alpha - \varepsilon$, then $G(t) \leq \alpha$ and vice versa.

- If $\hat{G}_n^{(j)}(t) > \alpha + \varepsilon$, then $G(t) > \alpha$ and vice versa.

It follows that the integrand in (21) is equal to 1 only if $\alpha - \varepsilon < \hat{G}_n^{(j)}(t) \leq \alpha + \varepsilon$ or $\alpha - \varepsilon < G(t) \leq \alpha + \varepsilon$, and equal to 0 otherwise. Again since $j \notin \mathcal{B}_d$, observe that

$$
\alpha - \varepsilon < \hat{G}_n^{(j)}(t) \leq \alpha + \varepsilon \implies \alpha - 2\varepsilon < G(t) \leq \alpha + 2\varepsilon.
$$

Thus, the integrand in (21) is 1 only if $\alpha - 2\varepsilon < G(t) \leq \alpha + 2\varepsilon$. This implies that

$$
|\mathbb{P}_{T^*}(\hat{G}_n^{(j)}(T^*) \leq \alpha) - \mathbb{P}_{T^*}(G(T^*))| \leq \mathbb{P}_{T^*}(\alpha - 2\varepsilon < G(T^*) \leq \alpha + 2\varepsilon).
$$

Observe that the set $\{t : \alpha - 2\varepsilon < G(t) \leq \alpha + 2\varepsilon\}$ has measure at most $4\varepsilon$ with respect to the marginal distribution of the core group survival times. Since $T^*$ is absolutely continuous with respect to the core group marginal survival time distribution and with bounded density, it follows that

$$
\sup_\alpha |\mathbb{P}_{T^*}(\hat{G}_n(T^*) \leq \alpha) - \mathbb{P}_{T^*}(G(T^*) \leq \alpha)| \leq \sup_\alpha \mathbb{P}_{T^*}(\alpha - 2\varepsilon < G(T^*) \leq \alpha + 2\varepsilon) \leq 4C\varepsilon.
$$

In particular, this means that the set of indices $j$ for which $\sup_\alpha |\mathbb{P}_{T^*}(\hat{G}_n^{(j)}(T^*) \leq \alpha) - \mathbb{P}_{T^*}(G(T^*) \leq \alpha)| > 4C\varepsilon$ is a subset of $\mathcal{B}_d$, so there are at most $\delta N + O(\sqrt{N})$ such indices.

Let $\sigma \in S_n$ be a permutation. Let $\mathbb{P}(\cdot \mid \sigma)$ denote the probability of an event conditional on $T(x_{\sigma(1)}) \leq \ldots \leq T(x_{\sigma(n)})$, and define $\sigma_j$ to be the permutation such that $t_{\sigma_j(1)}^{(j)} \leq \ldots \leq t_{\sigma_j(n)}^{(j)}$. Note that the total number of possible orderings of the survival times is $n!$, which is independent of $N$, and furthermore each permutation of the survival times occurs with positive probability independent of $N$.

Stepping back for a moment, recall that $\mathbb{P}_{\hat{G}_n, T^*}(\hat{G}_n(T^*) \leq \alpha \mid \sigma)$ (the cdf of the conditional rank statistics) actually consists of $n+1$ jump discontinuities at $\alpha \in \{k/n\}_{k=0}^n$, corresponding to the $n+1$ possible ranks of $T^*$ among $T_{\sigma(i)}$. Since $n$ is fixed with respect to $N$, by the strong law of large numbers and a union bound, we will have that

$$\lim_{N\to\infty} \frac{\sum_{j=1}^N \mathbb{P}_{T^*}(\mathrm{rk}(T^*) = k \mid T_1^{(j)}, \ldots, T_n^{(j)}, \sigma)\mathbb{1}\{\sigma_j = \sigma\}}{\sum_{j=1}^N \mathbb{1}\{\sigma_j = \sigma\}} = \mathbb{P}_{\hat{G}_n, T^*}(\mathrm{rk}(T^*) = k \mid \sigma)$$

almost surely, and simultaneously for all $k = 1, \ldots, n+1$. Thus, for any $\alpha \in [0, 1]$, we have

$$
\begin{aligned}
\mathbb{P}_{\hat{G}_n, T^*}(\hat{G}_n(T^*) \leq \alpha \mid \sigma) &= \sum_{k=1}^{\lfloor \alpha n \rfloor + 1} \mathbb{P}_{\hat{G}_n, T^*}(\mathrm{rk}(T^*) = k \mid \sigma) \\
&= \sum_{k=1}^{\lfloor \alpha n \rfloor + 1} \lim_{N\to\infty} \frac{\sum_{j=1}^N \mathbb{P}_{T^*}(\mathrm{rk}(T^*) = k \mid T_1^{(j)}, \ldots, T_n^{(j)}, \sigma)\mathbb{1}\{\sigma_j = \sigma\}}{\sum_{j=1}^N \mathbb{1}\{\sigma_j = \sigma\}} \\
&= \lim_{N\to\infty} \frac{\sum_{j=1}^N \sum_{k=1}^{\lfloor \alpha n \rfloor + 1} \mathbb{P}_{T^*}(\mathrm{rk}(T^*) = k \mid T_1^{(j)}, \ldots, T_n^{(j)}, \sigma)\mathbb{1}\{\sigma_j = \sigma\}}{\sum_{j=1}^N \mathbb{1}\{\sigma_j = \sigma\}} \\
&= \lim_{N\to\infty} \frac{\sum_{j=1}^N \mathbb{P}_{T^*}(\hat{G}_n^{(j)}(T^*) \leq \alpha)\mathbb{1}\{\sigma_j = \sigma\}}{\sum_{j=1}^N \mathbb{1}\{\sigma_j = \sigma\}}.
\end{aligned}
\tag{22}
$$

Let $\hat{p}_j = \hat{p}_j(\alpha) = \mathbb{P}(\hat{G}_n^{(j)}(T(x^*)) \leq \alpha)$ and $p^* = p^*(\alpha) = \mathbb{P}(F(T(x^*)) \leq \alpha)$. (We drop the $\alpha$ argument in what follows for clarity of presentation.) The preceding argument shows that $(\sum_{j=1}^N \hat{p}_j\mathbb{1}\{\sigma_j = \sigma\})/(\sum_{j=1}^N \mathbb{1}\{\sigma_j = \sigma\})$ converges to $\mathbb{P}_{\hat{G}_n, T^*}(\hat{G}_n(T^*) \leq \alpha \mid \sigma)$ uniformly in $\alpha$ as $N \to \infty$ and for all $\sigma$. It thus suffices to show that these finite sum estimates in (22) are sufficiently close to $p^*$ (uniformly in $\alpha$) with high probability.

To this end, we want to know for which permutations $\sigma$ can equation (22) deviate from $p^*$ by more than $4C\varepsilon + \eta$ for some small $\eta$ and for $N$ large enough. Let $N_\sigma = |\{j : \sigma_j = \sigma\}|$ and $N_\sigma^\times = |\{j : \sigma_j = \sigma \wedge \|\hat{p}_j - p^*\|_\infty > 4C\varepsilon\}|$. Observe that deterministically $\|\hat{p}_j - p^*\|_\infty \leq 1$, thus we have

$$
\begin{aligned}
\sup_\alpha \left| \frac{\sum_{j=1}^N \hat{p}_j\mathbb{1}\{\sigma_j = \sigma\}}{\sum_{j=1}^N \mathbb{1}\{\sigma_j = \sigma\}} - p^* \right| \\
\leq \frac{\sum_{j : \|\hat{p}_j - p^*\|_\infty \leq 4C\varepsilon, \sigma_j = \sigma} \|\hat{p}_j - p^*\|_\infty + \sum_{j : \|\hat{p}_j - p^*\|_\infty > 4C\varepsilon, \sigma_j = \sigma} \|\hat{p}_j - p^*\|_\infty}{N_\sigma} \\
\leq \frac{(N_\sigma - N_\sigma^\times)4C\varepsilon + N_\sigma^\times}{N_\sigma} \\
\leq 4C\varepsilon + \frac{N_\sigma^\times}{N_\sigma}.
\end{aligned}
$$

In particular, we would need $4C\varepsilon + \eta \leq 4C\varepsilon + N_\sigma^\times/N_\sigma$, which implies

$$N_\sigma^\times \geq \eta N_\sigma \geq \eta \mathbb{P}(\sigma)N - O(\sqrt{N})$$

with high probability. However, recall that the total number of indices $j$ with $|\hat{p}_j - p^*| > 4C\varepsilon$ is at most $\delta N + O(\sqrt{N})$, so

$\sum_\sigma N_\sigma^\times \leq \delta N + O(\sqrt{N})$. But then we have

$$\delta N + O(\sqrt{N}) \geq \sum_\sigma N_\sigma^\times$$
$$\geq \sum_{\sigma \in \mathcal{B}_\sigma} N_\sigma^\times$$
$$\geq \sum_{\sigma \in \mathcal{B}_\sigma} \eta \mathbb{P}(\sigma) N - O(\sqrt{N}).$$

Rearranging, we see that

$$\sum_{\sigma \in \mathcal{B}_\sigma} \mathbb{P}(\sigma) \leq \frac{\delta}{\eta} + O\left(\frac{1}{\sqrt{N}}\right). \tag{23}$$

In particular, we can take $\eta = \delta^{1/2}$.

To recap, we have shown that for any $\varepsilon, \delta > 0$, we can choose $n$ large enough such that $\mathbb{P}(\{x_i\}_{i=1}^n \in \mathcal{B}_x^n) < \delta$, and such that $\mathbb{P}(\sigma \in \mathcal{B}_\sigma \mid \{x_i\} \notin \mathcal{B}_x^n) \leq 2\delta^{1/2}$. (The factor of 2 is to absorb the $O(1/\sqrt{N})$ term in (23).) It follows that

$$\mathbb{P}\left\{\sup_\alpha \left|\mathbb{P}(\hat{G}_n(T(x^*)) \leq \alpha \mid \sigma) - \mathbb{P}(G(T^*)) \leq \alpha)\right| > 4C\varepsilon + \delta^{1/2}\right\}$$
$$\leq \mathbb{P}(\{x_i\}_{i=1}^n \in \mathcal{B}_x^n) + \mathbb{P}(\sigma \in \mathcal{B}_\sigma \mid \{x_i\}_{i=1}^n \notin \mathcal{B}_x^n)$$
$$\leq \delta + 2\delta^{1/2}.$$

As both the size of the error $4C\varepsilon + \delta^{1/2}$ and the failure probability $\delta + 2\delta^{1/2}$ can be made arbitrarily close to 0 by letting $\varepsilon, \delta \to 0$, we have that $\mathbb{P}(\hat{G}_n(T(x^*)) \leq \alpha | \sigma) \xrightarrow{p} \mathbb{P}(G(T(x^*)) \leq \alpha)$ uniformly in $\alpha$, as desired. $\quad\square$

## F. Proof of Theorem 6.1

### F.1. Assumptions and Notation

We first restate our assumptions for convenience.

A1 The hazard function for the entire dataset has the form $\lambda(t; x) = \lambda_0(t)e^{h(x)}$ for some unknown risk function $h$.

A2 There is no censoring in the data.

A3 There is a unique largest region $R^*$ which minimizes the EPE, and $R^*$ is an axis-aligned box. Conditional on $x \in R^*$, we have $h(x) = \beta^\top x$ for some $\beta$, i.e., the Cox model is well-specified.

A4 The core group selection procedure finds a group of points which belong to $R^*$, and the Cox model fit to these points recovers the true parameters $\beta$.

A5 The error between the finite conditional rank statistics and its large-sample limit according to Theorem E.1 is negligible. We use this limiting distribution for the analysis, rather than the finite sample version described in Section 4.

We define the following quantities:

- $T(x)$ denotes the survival time for a point with features $x$, i.e., for each datapoint $x_i$ in the dataset, $t_i = T(x_i)$. By our assumptions on the data generating distribution, $T(x)$ is a survival time with hazard function $\lambda(t, x) = \lambda_0(t)e^{h(x)}$.

- $G(t; R_{\text{core}})$ denotes the marginal CDF for survival times sampled from points belonging to the core group. That is, $G(t) = \mathbb{P}(T(X) \leq t \mid X \in R_{\text{core}})$, where the probability is computed with respect to both the randomness in $X \in R_{\text{core}}$ and $T(X)$. To simplify the notation, we will typically write $G(t) = G(t; R_{\text{core}})$.

- $G^{-1} : [0, 1) \to \mathbb{R}_{\geq 0}$ denotes the inverse CDF.

- $p_{\text{I}}$ denotes the type I error rate, i.e., the probability that a point which belongs to $R^*$ is rejected.

- $p_{\text{II}}$ denotes the type II error rate, i.e., the probability that we fail to reject a point close to each face of $R^*$.

- $2n$ denotes the total number of datapoints in the training dataset, $n$ of which are used to find the core group and $n$ of which are used for the rejection/expansion step. We split the dataset to avoid creating dependencies between the datapoints after the core group selection step.

The necessity and validity of these assumptions is discussed in Appendix G.

### F.2. Theoretical Implementation of DDGroup

We use a modified version of the algorithm for the proof of Theorem 6.1. Compared to the practical implementation, the main differences are:

- Using the large-sample limit $G$ of the CRS to reject points as opposed to the finite sample version (Assumption A5);

- Using approximate, rather than exact, quantiles of $G$ to reject points;

- Splitting the training data so that disjoint subsets are used for finding the core group and performing the rejection/expansion step to avoid introducing intricate dependencies between the survival times (the practical implementation uses the entire training set for both steps);

- Assuming $\hat{\beta} = \beta$ when fit to the core group (Assumption A4); and

- The inclusion of an additional hyperparameter, the expansion speed of the sides of the growing box, as in the theory for the original DDGroup (Izzo et al., 2023).

Several of the following descriptions are replicated (nearly) verbatim from (Izzo et al., 2023) for the reader's convenience.

Let $U \subseteq \mathbb{R}^d$. We define the *directed infinity norm* $\|x\|_{U,\infty}$ by

$$\|x\|_{U,\infty} = \max_{u \in U} x^\top u.$$

We note that for many sets $U$, $\|\cdot\|_{U,\infty}$ may not be a norm, nor even a seminorm. In what follows, $U$ will initially be defined as $U = \{\pm s_j^\pm e_i\}_{i=1}^d$. When $s_j^\pm = 1$ for all dimensions $j$ and signs $\pm$, $\|\cdot\|_{U,\infty} = \|\cdot\|_\infty$ coincides with the usual infinity norm on $\mathbb{R}^d$. With generic "speed" parameters $s_j^\pm$, this quantity corresponds to the $\ell_\infty$ norm after first rescaling the positive/negative axes by $1/s_j^+$ and $1/s_j^-$, respectively.

---

**Algorithm 2** CoreGroup($S_{\text{core}}, D$)

---

**Require:** Core group shape $S_{\text{core}}$, dataset $D = \{(x_i, t_i)\}_{i=1}^n$
    $\text{EPE}^* \leftarrow \infty$
    **for** $(x, t) \in D$ **do**
        $D_{\text{nbhd}} \leftarrow \{(x', t') \in D \mid x' - x \in S_{\text{core}}\}$ {$D_{\text{nbhd}}$ contains all points in a set of shape $S_{\text{core}}$ centered at $x$. For instance, $S_{\text{core}}$ could be an $\ell_\infty$ ball of radius $r$; then $D_{\text{nbhd}}$ contains all points within an $\ell_\infty$-distance $r$ from $x$.}
        $\hat{\beta} \leftarrow \text{FitCox}(D_{\text{nbhd}})$ {Fit the Cox model using the log partial likelihood.}
        **if** $\text{EPE}(\hat{\beta}, D_{\text{nbhd}}) < \text{EPE}^*$ **then**
            $D_{\text{core}} \leftarrow D_{\text{nbhd}}$
            $R_{\text{core}} \leftarrow x + S_{\text{core}}$
            $\text{EPE}^* \leftarrow \text{EPE}(\hat{\beta}, D_{\text{nbhd}})$
        **end if**
    **end for**
    **return** $R_{\text{core}}, D_{\text{core}}$

---

---

**Algorithm 3** GROWBOX($\bar{x}, X_{\text{rej}}, \{s_j^{\pm}\}_{j=1}^d$)

---

**Require:** Starting point (center) $\bar{x}$, rejected points $X_{\text{rej}}$, side expansion speeds $\{s_j^{\pm}\}_{j=1}^d$

$\hat{R} \leftarrow \emptyset$

$U \leftarrow \{-s_j^- e_j, s_j^+ e_j\}_{j=1}^d$ {$e_j$ denotes the $j$-th standard basis vector.}

$X_{\text{rej}} \leftarrow X_{\text{rej}} + \{-\bar{x}\}$ {Center the points at $\bar{x}$. $+$ denotes Minkowski sum.}

**while** $X_{\text{rej}} \neq \emptyset$ **do**

    $x^* \leftarrow \text{argmin}_{x \in X_{\text{rej}}}\{\|x\|_{U,\infty}\}$

    $a^* \leftarrow \|x^*\|_{U,\infty}$

    $u^* \leftarrow \text{argmax}_{u \in U}\{u^\top x^*\}$ {$u^*$ is the next support direction for the polytope}

    Add $(u^*, a^*)$ to $\hat{R}$

    Remove $u^*$ from $U$

    $X_{\text{rej}} \leftarrow \{x \in X_{\text{rej}} \mid x^\top u^* < a^*\}$

**end while**

**return** $\hat{R} + \{\bar{x}\}$ {Undo the centering procedure from the first part of the algorithm.}

---

**Algorithm 4** DDGROUP($R_{\text{core}}, \{s_j^{\pm}\}_{j=1}^d, D$)

---

**Require:** Core group shape $S_{\text{core}}$, growth speeds $s_j^{\pm}$, dataset $D = \{(x_i, t_i)\}_{i=1}^{2n}$

**Phase 1:** Find a core group and fit a coarse model using half of the data.

$R_{\text{core}}, D_{\text{core}} \leftarrow \text{CORE GROUP}(S_{\text{core}}, \{x_i\}_{i=n+1}^n)$

$\hat{\beta} \leftarrow \text{FITCOX}(D_{\text{core}})$ {This fits the Cox model by minimizing the log partial likelihood. We assume $\hat{\beta} = \beta$ (Assumption A4)}

**Phase 2:** Label which points should be excluded using the other half of the training data.

**for** $i = 1, \ldots, n$ **do**

    $\ell_i \leftarrow \mathbb{1}\{G(t_i; R_{\text{core}}) \notin [\underline{q}(x), \overline{q}(x)]\}$ {The lower and upper quantiles $\underline{q}, \overline{q}$ are defined in Lemma F.3.}

**end for**

$X_{\text{rej}} \leftarrow \{x_i \in X \mid \ell_i = 1\}$

**Phase 3:** Approximate $R^*$.

$\bar{x} \leftarrow \text{MEAN}(\{x \mid (x, t) \in D_{\text{core}}\})$

$\hat{R} \leftarrow \text{GROWBOX}(\bar{x}, X_{\text{rej}}, \{s_j^{\pm}\}_{j=1}^d)$

**return** $\hat{R}$

---

### F.3. Proof

We now proceed with the proof of Theorem 6.1, which is broken down into five steps.

**Lemma F.1.** *We may assume WLOG that $\lambda_0(t) \equiv 1$.*

*Proof.* Consider the random variable $\tilde{T} = \int_0^T \lambda_0(s)\,ds$. Note that this is a monotonic change of variables since $\lambda_0(s) > 0$. In addition, note that the survival function $\tilde{S}(t, x)$ of $\tilde{T}$ conditional on features $X = x$ is given by

$$
\begin{aligned}
\tilde{S}(t, x) &= \mathbb{P}(\tilde{T} \geq t \mid X = x) \\
&= \mathbb{P}\left(\int_0^T \lambda_0(s)\,ds \geq t \mid X = x\right) \\
&= \mathbb{P}(T \geq \Lambda_0^{-1}(t) \mid X = x) \\
&= \exp\left(-e^{h(x)}\Lambda_0(\Lambda_0^{-1}(t))\right) \\
&= \exp(-e^{h(x)}t).
\end{aligned}
\tag{24}
$$

This implies that $\tilde{T}$ has hazard function $\tilde{\lambda}(t; x) = e^{h(x)}$, which in particular means that the baseline hazard function under this transformation is $\tilde{\lambda}_0(t) \equiv 1$. Since the transformation is monotonic, all of the ranks will be preserved, so all of the results which hold for $T$ hold also for $\tilde{T}$ and vice-versa. $\qquad \square$

**Lemma F.2.** *Let $p$ be an upper bound on the probability that a point $x \in R^*$ is rejected. If $p \leq p_\mathrm{I}/n$, then the probability that* any *point in $R^*$ is rejected is at most $p_\mathrm{I}$.*

*Proof.* Since the size of the dataset is $n$, there are clearly at most $n$ points in $R^*$. The result then follows from a simple union bound over these points. $\qquad \square$

**Lemma F.3.** *Define the lower and upper rejection quantiles*

$$\underline{q}(x) = 1 - \mathbb{E}_{X \sim \mathrm{Core}}\left[\left(1 - \frac{p}{2}\right)^{e^{\beta^\top(X-x)}}\right], \qquad \overline{q}(x) = 1 - \mathbb{E}_{X \sim \mathrm{Core}}\left[\left(\frac{p}{2}\right)^{e^{\beta^\top(X-x)}}\right]$$

*respectively. Using these quantiles, the probability of incorrectly rejecting a point $x \in R^*$ is at most $p$, i.e.,*

$$\mathbb{P}(G(T(x)) \notin [\underline{q}(x), \overline{q}(x)]) \leq p.$$

*Proof.* By a union bound, it suffices to choose $\underline{q}(x), \overline{q}(x)$ such that

$$\mathbb{P}(G(T(x)) < \underline{q}(x)) \leq p/2, \qquad \mathbb{P}(G(T(x)) > \overline{q}(x)) \leq p/2.$$

Let us start with the lower quantile. We will abbreviate $\underline{q} = \underline{q}(x)$. Given $x$, we wish to determine the maximum possible $\underline{q}$ such that

$$\mathbb{P}(T(x) \leq G^{-1}(\underline{q})) \leq \frac{p}{2} \quad \Leftrightarrow \quad \mathbb{P}(T(x) > G^{-1}(\underline{q})) \geq 1 - \frac{p}{2}.$$

From Lemma F.1, it suffices to consider the case where $\lambda_0(t) \equiv 1$. In this case, since $x \in R^*$, $T(x)$ is an exponential random variable with rate $e^{\beta^\top x}$ and we can compute the tail probability explicitly. In particular, we have

$$\mathbb{P}(T(x) > G^{-1}(\underline{q})) = \exp(-e^{\beta^\top x}G^{-1}(\underline{q})) \geq 1 - \frac{p}{2} \tag{25}$$

$$\Leftrightarrow \quad -e^{\beta^\top x}G^{-1}(\underline{q}) \geq \log(1 - \frac{p}{2}) \tag{26}$$

$$\Leftrightarrow \quad G^{-1}(\underline{q}) \leq -\log(1 - \frac{p}{2})e^{-\beta^\top x} \tag{27}$$

$$\Leftrightarrow \quad \underline{q} \leq G\left(-\log(1 - \frac{p}{2})e^{-\beta^\top x}\right) \tag{28}$$

$$= \mathbb{P}_{T \sim \mathrm{Core}}\left(T \leq -\log(1 - \frac{p}{2})e^{-\beta^\top x}\right) \tag{29}$$

$$= \mathbb{E}_{X \sim \mathrm{Core}}\left[1 - \exp\left(-e^{\beta^\top X} \cdot (-\log(1 - \frac{p}{2})e^{-\beta^\top x})\right)\right] \tag{30}$$

$$= 1 - \mathbb{E}_{X \sim \mathrm{Core}}\left[(1 - \frac{p}{2})^{e^{\beta^\top(X-x)}}\right]. \tag{31}$$

In particular, we can take $\underline{q}(x)$ equal to the expression in (31). A similar calculation for the upper quantile yields the expression for $\overline{q}(x)$. $\qquad \square$

**Lemma F.4.** *Let $\{\tilde{x}_i\}_{i=1}^m$ be $m$ points with independent survival times be such that*

$$\mathbb{P}(\tilde{x}_i \text{ is rejected}) \geq \eta$$

*for all $i$. If $\eta \geq 1 - p_\mathrm{II}^{1/m}$, then the probability that none of the $x_i$ are rejected is at most $p_\mathrm{II}$.*

*Proof.* By the independence of the survival times, the probability that none of the points are rejected is at most

$$(1 - \eta)^m \leq (1 - (1 - p_{\mathrm{II}}^{1/m}))^m = p_{\mathrm{II}}$$

as desired. □

**Lemma F.5.** *Suppose $\{x_i\}_{i=1}^n$ are drawn i.i.d. from the uniform distribution on $\mathcal{B} \subseteq \mathbb{R}^d$ with $\mathrm{vol}(\mathcal{B}) = B < \infty$. Let $S \subseteq \mathcal{B}$ be any region such that $\mathrm{vol}(S) \geq C > 0$. Then $|\{x_i \mid x_i \in S\} \geq \frac{Cn}{2B}$ with probability at least $1 - 1/n$ for large enough $n$.*

*Proof.* Since the $x_i$ are drawn from the uniform distribution, we have that $\mathbb{P}(x_i \in S) \geq C/B$. This means that $\mathbb{1}\{x_i \in S\} \sim \mathrm{Bernoulli}(q)$ with $q \geq C/B$. Thus by Hoeffding's inequality, with probability at least $1 - 1/n$ we have that

$$|\{x \; : \; |u^\top x| < r\}| \geq qn - \sqrt{\frac{n \log n}{2}} \geq \frac{Cn}{2B}$$

for large enough $n$. □

**Lemma F.6.** *There exists $C_\varepsilon = O(\log \varepsilon^{-1})$ such that if $|h(x) - \beta^\top x| \geq C_\varepsilon$, then the probability of rejecting $x$ is at least $\eta$:*

$$\mathbb{P}(G(T(x)) \notin [\underline{q}(x), \, \overline{q}(x)]) \geq \eta.$$

*Proof.* By Lemma F.1, it suffices to consider $\lambda_0(t) \equiv 1$ so that $T(x)$ is exponential with rate $e^{h(x)}$. We will use this fact throughout the proof. We will also make repeated use of the fact that $G$ and $G^{-1}$ are monotonically increasing functions without stating this explicitly.

By Lemma F.7, we have that $-\log^2(\frac{1}{1-p/2}) + \log^2(\frac{1}{1-\eta})$ and $\log^2(2/p) - \log^2(1/\eta)$ are both $O(\log \varepsilon^{-1})$. Thus, it suffices to show that if

$$h(x) \leq \beta^\top x - \log^2 \frac{2}{p} + \log^2 \frac{1}{\eta} \quad \text{or} \quad h(x) \geq \beta^\top x - \log^2(\frac{1}{1-p/2}) + \log^2(\frac{1}{1-\eta})$$

then the rejection probability is at least $\eta$ for $x$. The value of $C_\varepsilon$ can then be chosen as the $O(\log \varepsilon^{-1})$ upper bound for these two quantities.

We will now consider two cases based on whether $h(x) > \beta^\top x$ or $h(x) < \beta^\top x$. In the former case, we use the fact that

$$\mathbb{P}(G(T(x)) \notin [\underline{q}(x), \, \overline{q}(x)]) \geq \mathbb{P}(G(T(x)) < \underline{q}(x))$$

and lower bound the RHS; similarly, when $h(x) < \beta^\top x$, we use the fact that

$$\mathbb{P}(G(T(x)) \notin [\underline{q}(x), \, \overline{q}(x)]) \geq \mathbb{P}(G(T(x)) > \overline{q}(x))$$

and lower bound the RHS.

Let us first suppose $h(x) > \beta^\top x$. Let $\underline{q} = \underline{q}(x)$. Then we have

$$\mathbb{P}(G(T(x)) \leq \underline{q}) \geq \eta \quad \Leftrightarrow \quad \mathbb{P}(T(x) > G^{-1}(\underline{q})) \leq 1 - \eta \tag{32}$$

$$\Leftrightarrow \quad \exp(-e^{h(x)} G^{-1}(\underline{q})) \leq 1 - \eta \tag{33}$$

$$\Leftrightarrow \quad e^{h(x)} G^{-1}(\underline{q}) \geq -\log(1 - \eta) \tag{34}$$

$$\Leftrightarrow \quad h(x) \geq \log^2\left(\frac{1}{1-\eta}\right) - \log G^{-1}(\underline{q}). \tag{35}$$

Using the expression for $\underline{q} = \underline{q}(x)$ from Step 3, for any $t \geq 0$, we have

$$\mathbb{P}_{T \sim \text{Core}}(T \leq t) \leq \underline{q} \tag{36}$$

$$\Leftrightarrow \quad \mathbb{E}_{X \sim \text{Core}}\left[1 - \exp(-e^{\beta^\top X} t)\right] \leq 1 - \mathbb{E}_{X \sim \text{Core}}\left[(1 - \frac{p}{2})^{e^{\beta^\top (X - x)}}\right] \tag{37}$$

$$\Leftrightarrow \quad \mathbb{E}_{X \sim \text{Core}}\left[\exp\left(\log(1 - \frac{p}{2}) e^{-\beta^\top x} e^{\beta^\top X}\right)\right] \leq \mathbb{E}_{X \sim \text{Core}}\left[\exp(-t e^{\beta^\top X})\right]. \tag{38}$$

As long as $t \leq -\log(1 - \frac{p}{2}) e^{-\beta^\top x}$, the integrand on the LHS of (38) is pointwise less than or equal to the integrand on the RHS. In particular, with $t$ equal to this upper bound we have $G(t) = \mathbb{P}_{T \sim \text{Core}}(T \leq t) \leq \underline{q}$, which implies that

$$G^{-1}(\underline{q}) \geq -\log(1 - \frac{p}{2}) e^{-\beta^\top x} \quad \implies \quad -\log G^{-1}(\underline{q}) \leq \beta^\top x - \log^2\left(\frac{1}{1 - p/2}\right).$$

Thus we have

$$h(x) \geq \beta^\top x + \log^2\left(\frac{1}{1 - \eta}\right) - \log^2\left(\frac{1}{1 - p/2}\right) \geq \log^2\left(\frac{1}{1 - \eta}\right) - \log G^{-1}(\underline{q}). \tag{39}$$

In particular, inequality (35) is satisfied, which implies that $\mathbb{P}(G(T(x)) \leq \underline{q}(x)) \geq \eta$ as desired.

Otherwise, suppose that $h(x) < \beta^\top x$. Now we have

$$\mathbb{P}(G(T(x)) > \overline{q}) \geq \eta \quad \Leftrightarrow \quad \exp(-e^{h(x)} G^{-1}(\overline{q})) \geq \eta \tag{40}$$

$$\Leftrightarrow \quad e^{h(x)} G^{-1}(\overline{q}) \leq -\log \eta \tag{41}$$

$$\Leftrightarrow \quad h(x) \leq \log^2\left(\frac{1}{\eta}\right) - \log G^{-1}(\overline{q}). \tag{42}$$

In order to lower bound $-\log G^{-1}(\overline{q})$, we wish to establish conditions on $t$ such that $G(t) = \mathbb{P}_{T \sim \text{Core}}(T \leq t) \geq \overline{q}$. Using the expression from ..., we have

$$\mathbb{P}_{T \sim \text{Core}}(T \leq t) \geq \overline{q} \tag{43}$$

$$\Leftrightarrow \quad \mathbb{E}_{X \sim \text{Core}}\left[1 - \exp(-e^{\beta^\top X} t)\right] \geq 1 - \mathbb{E}_{X \sim \text{Core}}\left[(\frac{p}{2})^{e^{\beta^\top (X - x)}}\right] \tag{44}$$

$$\Leftrightarrow \quad \mathbb{E}_{X \sim \text{Core}}\left[\exp\left(\log(\frac{p}{2}) e^{-\beta^\top x} e^{\beta^\top X}\right)\right] \geq \mathbb{E}_{X \sim \text{Core}}\left[\exp(-t e^{\beta^\top X})\right]. \tag{45}$$

Again, inequality (45) can be made to hold pointwise in the integrand provided that $t \geq -\log(\frac{p}{2}) e^{-\beta^\top x}$. It therefore follows that

$$G^{-1}(\overline{q}) \leq -\log(\frac{p}{2}) e^{-\beta^\top x} \quad \implies \quad -\log G^{-1}(\underline{q}) \geq \beta^\top x - \log^2\left(\frac{2}{p}\right)$$

Finally, we have

$$h(x) \leq \beta^\top x - \log^2\left(\frac{2}{p}\right) + \log^2\left(\frac{1}{\eta}\right) \leq \log^2\left(\frac{1}{\eta}\right) - \log G^{-1}(\overline{q}).$$

Thus (42) holds and therefore $\mathbb{P}(G(T(x)) \geq \overline{q}(x)) \geq \eta$ as desired. $\qquad \square$

To prove the desired result, we can now directly apply logic from the analogous proof in (Izzo et al., 2023). Specifically, since we have assumed that the core group lies within $R^*$ and we have shown that no points in $R^*$ will be rejected (Step 2), the logic of (Izzo et al., 2023) shows that $R^* \subseteq \hat{R}_N$ given correct settings for the expansion speed $s_j^\pm$ of each side of the box. Similarly, the choice of $C_\varepsilon$ implies that there will be a rejected point within an $O(\varepsilon)$ distance from each face of $R^*$ (Steps 4 & 5), meaning that each face of $\hat{R}_N$ will stop expanding within $O(\varepsilon)$ distance of the corresponding face of $R^*$ and yielding $\text{vol}(\hat{R}_N \setminus R^*) = O(\varepsilon)$. This completes the proof.

**Lemma F.7.** *Let $p = p_{\mathrm{I}}/n$, $\eta = 1 - p_{\mathrm{II}}^{1/m}$, and $m \geq c\varepsilon n$. Then as $n \to \infty$, we have*

$$-\log^2\left(\frac{1}{1-p/2}\right) + \log^2\left(\frac{1}{1-\eta}\right) = O(\log\varepsilon^{-1}),$$

$$\log^2\left(\frac{2}{p}\right) - \log^2\left(\frac{1}{\eta}\right) = o(1) = O(\log\varepsilon^{-1}).$$

*Proof.* We begin with the first bound. Substituting for $p$, we have that

$$-\log\log\frac{1}{1-p_{\mathrm{I}}/2n} \leq -\log\log(1 + p_{\mathrm{I}}/2n) \leq -\log\frac{p_{\mathrm{I}}}{4n}$$

for large enough $n$. (Here we have used $\log(1+z) \geq z/2$ for small $z > 0$.) Substituting for $\eta$ and $m$, we also have

$$\log\log\frac{1}{1-\eta} \leq \log\left(\frac{1}{m}\log p_{\mathrm{II}}^{-1}\right) \leq -\log(c\varepsilon n) + \log^2 p_{\mathrm{II}}^{-1}.$$

It therefore follows that

$$\begin{aligned}
-\log^2\left(\frac{1}{1-p/2}\right) + \log^2\left(\frac{1}{1-\eta}\right) &\leq \log\frac{4n}{p_{\mathrm{I}}} - \log(c\varepsilon n) + \log^2 p_{\mathrm{II}}^{-1} \\
&= \log\varepsilon^{-1} + \log(4cp_{\mathrm{I}}^{-1}) + \log^2 p_{\mathrm{II}}^{-1} \\
&= O(\log\varepsilon^{-1}).
\end{aligned}$$

We now turn to the second bound. First, observe that $p_{\mathrm{II}}^{1/m} = e^{\frac{1}{m}\log p_{\mathrm{II}}} \geq 1 + \frac{1}{m}\log p_{\mathrm{II}}$. We therefore have that

$$-\log^2\left(\frac{1}{\eta}\right) \leq -\log^2\frac{1}{1-p_{\mathrm{II}}^{1/m}} \leq -\log^2\frac{m}{\log p_{\mathrm{II}}^{-1}} \leq -\log^2\frac{c\varepsilon n}{\log p_{\mathrm{II}}^{-1}}.$$

Thus, it follows that

$$\log^2\left(\frac{2}{p}\right) - \log^2\left(\frac{1}{\eta}\right) \leq \log^2\frac{2n}{p_{\mathrm{I}}} - \log^2\frac{c\varepsilon n}{\log p_{\mathrm{II}}^{-1}}.$$

We can therefore apply Lemma F.8 with $c_1 = 2/p_{\mathrm{I}}$ and $c_2 = c\varepsilon/\log(p_{\mathrm{II}}^{-1})$ to conclude that $\log^2\left(\frac{2}{p}\right) - \log^2\left(\frac{1}{\eta}\right) \to 0$. In particular, this term is also $O(\log\varepsilon^{-1})$. $\qquad\square$

**Lemma F.8.** *For any positive constants $c_1, c_2 > 0$, we have $\lim_{n\to\infty}\log^2(c_1 n) - \log^2(c_2 n) = 0$.*

*Proof.* This is a straightforward calculation:

$$\log^2(c_1 n) - \log^2(c_2 n) = \log\left(\frac{\log(c_1 n)}{\log(c_2 n)}\right) = \log\left(\frac{\log c_1 + \log n}{\log c_2 + \log n}\right) = \log\left(1 + \frac{\log c_1 - \log c_2}{\log c_2 + \log n}\right) \to \log(1+0) = 0.$$

Note that we have used the continuity of the logarithm. $\qquad\square$

**Theorem 6.1.** *Let $\hat{R}_n$ be the region output by Algorithm 4 on a dataset of $2n$ i.i.d. points satisfying Assumptions A1-A5. For any $\varepsilon > 0$, there is an effect size $C_\varepsilon = O(\log\varepsilon^{-1})$ such that if $|h(x) - \beta^\top x| \geq C_\varepsilon$ outside of $R^*$, then there exist settings for the hyperparameters of Alg. 4 such that with probability at least 0.99, $R^* \subseteq \hat{R}_n$ and $\mathrm{vol}(\hat{R}_n \setminus R^*) \leq C'\varepsilon$ for another constant $C'$ as $n \to \infty$.*

*Proof.* Let $p = p_{\mathrm{I}}/n$. By Lemma F.3, the probability that any point $x \in R^*$ is rejected is at most $p$. Since there are at most $n$ points in $R^*$ (as the total size of the dataset is $n$), by Lemma F.2, the probability that any point in $R^*$ is rejected is at most $p_{\mathrm{I}}$.

Since $R^*$ is an axis-aligned box, we can write $R^* = \prod_{j=1}^{d}[a_j, b_j]$. For any vector $x \in \mathbb{R}^d$, let $x[k]$ denote the $k$-th entry of $x$. As in (Izzo et al., 2023), for each $j = 1, \ldots, d$, we define

$$\partial R^*_{\varepsilon,j,+} = \{x \in \mathbb{R}^d \mid b_j \leq x[j] \leq b_j + \varepsilon,\ a_k \leq x[k] \leq b_k, k \neq j\},$$

$$\partial R^*_{\varepsilon,j,-} = \{x \in \mathbb{R}^d \mid a_j - \varepsilon \leq x[j] \leq a_j,\ a_k \leq x[k] \leq b_k, k \neq j\}.$$

These are the sets of points which are at most $\varepsilon$ "above" the upper dimension $j$ face of $R^*$ and "below" the lower dimension $j$ face of $R^*$, respectively.

Fix a dimension $j$ and let $m$ be the number of points in $\partial R^*_{\varepsilon,j,+}$. By Lemma F.5, $m \geq c\varepsilon n$ for some constant $c > 0$ with probability at least $1 - 1/n$. Let $\eta = 1 - p_{\mathrm{II}}^{1/m}$, and let $C_\varepsilon = O(\log \varepsilon^{-1})$ be as defined in Lemma F.7. Then by the result of Lemma F.7, we have that each of the $m$ points in $\partial R^*_{\varepsilon,j,+}$ has probability at least $\eta$ of being (correctly) rejected. By Lemma F.4, the probability that no points in $\partial R^*_{\varepsilon,j,+}$ are rejected is at most $p_{\mathrm{II}}$. The same logic clearly holds for $R^*_{\varepsilon,j,-}$ as well. By a union bound over $j = 1, \ldots, d$ and over the sign $\pm$, we have that all of the $\partial R^*_{\varepsilon,j,\pm}$ contain a rejected point with probability at least $1 - 2d \cdot (p_{\mathrm{II}} + 1/n)$.

By Assumption A4, the core group is a subset of $R^*$, hence its average $\bar{x}$ is contained in $R^*$.

For the remainder of the proof, we will condition on the event that $R^*$ contains no rejected points and all of the $\partial R^*_{\varepsilon,j,\pm}$ contain at least one rejected point. Again by a union bound, this occurs with probability at least

$$1 - p_{\mathrm{I}} - 2d \cdot (p_{\mathrm{II}} + 1/n).$$

We can now follow the logic of Theorem 4.3 of (Izzo et al., 2023) with almost no modification, which we replicate here for convenience. Let $s_j^{\pm} = d(\bar{x}, \partial R^*_{j,\pm})$ be the distance from the core group center to the appropriate face of $R^*$. Note that Algorithm 3 with these speeds and this center is equivalent to running the algorithm from the origin and with uniform speeds, after shifting the data so that $\bar{x}$ lies at the origin and then rescaling each positive and negative axis by $1/s_j^+$ and $1/s_j^-$, respectively. In this case, $R^*$ is transformed into a $\ell_\infty$ ball of radius 1 centered at the origin. We have also conditioned on the event that $R^*$ contains no rejected points, and the transformations we performed above preserve this fact. Since the region returned by Algorithm 3 returns a region which contains the largest centered $\ell_\infty$ ball with no rejected points in it, and $R^*$ is a centered $\ell_\infty$ ball with no rejected points, we must have $R^* \subseteq \hat{R}$ as desired.

Because each $\partial R^*_{\varepsilon,j,\pm}$ contains at least one rejected point, Algorithm 3 will stop growing the $(j, \pm)$ side of $\hat{R}$ at some point in $\partial R^*_{\varepsilon,j,\pm}$. In particular, this means that if the final region $\hat{R}_n = \prod_{j=1}^{d}[\ell_j, u_j]$ then we have

$$\ell_j \geq a_j - \varepsilon, \qquad u_j \leq b_j + \varepsilon$$

for all $j$. (If any of these inequalities were violated, then since $R^* \subseteq \hat{R}_n$, by convexity $\hat{R}_n$ would also have to contain a rejected point.) Thus we can conclude that $\mathrm{vol}(\hat{R}_n \setminus R^*) = O(\varepsilon)$. This completes the proof. $\qquad \square$

### F.4. Discussion of the Result

Regarding the convergence result itself, the natural point of comparison is with the convergence result of the version of DDGroup intended for use with linear regression in (Izzo et al., 2023). At first glance, the result of Theorem 6.1 may appear weaker: given a fixed effect size ($C_\varepsilon$) and sufficiently many samples, we recover $R^*$ up to a fixed $O(\varepsilon)$ error in volume. In contrast, (Izzo et al., 2023) obtains arbitrary precision given enough samples and at a fixed effect size. The reason for this disparity is that (Izzo et al., 2023) assumed a lower bound on the *variance* of $y|x$ outside of $R^*$; our assumption on the effect size is on the equivalent of the regression function. Since detections are performed based on the deviation of a single point, and the Gaussian fluctuations in (Izzo et al., 2023) grow as $\sigma \log n$, where $\sigma$ is the standard deviation and $n$ the number of datapoints, their convergence result would also require a effect size in the regression function which increases with the dataset size $n$ (at least $\Omega(\log n)$) in order to obtain arbitrary precision with increasing sample size. Indeed, the proof of Theorem 6.1 shows that $C_\varepsilon = O(\log \varepsilon^{-1})$, so to obtain an error of size $O(n^{-c})$ for some constant $c > 0$ (as is the case in (Izzo et al., 2023)), DDGroup adapted to the Cox model would also need an effect size of order $O(\log n)$. Thus, these two results are comparable in their recovery guarantees.

We also remark that the bounds on the effect size resulting from Lemma F.6 are not symmetric for the upper and lower tails. The effect size is determined by the amount which the parameter of an exponential distribution must be changed in order to

move its upper or lower tails to lie above or below (respectively) the upper and lower quantiles $\underline{q}(x)$ and $\overline{q}(x)$. Although $\underline{q}$ and $\overline{q}$ are defined as symmetric upper/lower quantiles of the marginal core group survival distribution, the size of change required to move the upper and lower tails of the exponential distribution past a certain point is not symmetric. Thus, we should not expect the effect size required for upper/lower rejections to be the same.

Because of the asymmetry in the upper and lower tails, it is true that we can improve (reduce) the required effect size by balancing the thresholds required for upper and lower tail rejections. It is possible it will even lead to asymptotic improvement with respect to $\varepsilon^{-1}$, but since the point of the theory is just to get a sense of the algorithm's performance, we do not explore this further here.

## G. Discussion of Assumptions

We briefly discuss the necessity and validity of the assumptions.

Relaxing Assumptions A1 (to include general hazard functions outside of $R^*$ and A2 (to include censoring) should be possible. As the present paper is the first to consider even the more basic forms of these questions, we leave these more challenging extensions to future work.

The existence of a "good" subgroup is dataset dependent. Assumption A3 guarantees that a "good" subgroup of the data exists, and we should only expect to have performance guarantees in such a setting.

Finally, Theorem E.1 guarantees that Assumption A5 is a reasonable approximation given sufficient data.

Assumption A4 is a stronger assumption at face value, so we discuss its necessity separately in the following subsections.

### G.1. Necessity of Assumption A4

We provide a negative result showing the necessity of Assumption A4, i.e., that the core group selection is contained in $R^*$. The problem arises because subsets of the best *overall* region do not necessarily obtain the minimum EPE among subsets of a fixed shape.

Consider the following counterexample. The features are supported on $\mathcal{B} = [-1, 1]$ and consist of three regions.

In $R^* = [-1, 0]$, $T|X$ follows a Cox model with moderate signal: $T|X$ has hazard function $\lambda(t, x) = e^{\beta_1 x}$ when $x \in R^*$.

In $R^+ = [1 - \varepsilon, 1]$, $T|X$ also follows a Cox model with a larger signal, but which does not compensate for the smaller region size: $\lambda(t, x) = e^{\beta_2 x}$ when $x \in R^+$ with $|\beta_2| > |\beta_1|$.

Finally, in $R^\circ = \mathcal{B} \setminus (R^* \cup R^+) = (0, 1 - \varepsilon)$, we have that $T|X \sim \text{Unif}(\{0, \infty\})$.

Figure 2 shows the EPE for all possible regions with a grid size of 0.1. For this example, we used $\varepsilon = 0.1$, $\beta_1 = -4$, and $\beta_2 = 25$ with $n = 4000$ points for the whole dataset with which to compute Monte Carlo estimates for the EPE of each region. As in Figure 1, the cell with bottom-left corner at $(a, b)$ corresponds to the EPE for the region $[a, b]$. The color of each cell denotes the EPE value, given by the color bar on the right of the plot.

Due to the combination of a favorable feature distribution (a larger region, making the units easier to distinguish) and a moderate signal conditional on the features, the minimum EPE is obtained by $R^* = [-1, 0]$, denoted by the cyan outlined square. The black outlined squares denote intervals of size 0.1. If we consider core groups of this size, then the minimum among such groups will be obtained by $R^+ = [0.9, 1]$, denoted by the red outlined square. As this is not a subset of $R^*$, even after the expansion step, we will not obtain the correct region.

We remark that an alternative version of Theorem 6.1 holds under a weaker version of Assumption A4. Specifically, if one assumes that the core group selection procedure returns a core group which is fully contained in *any* axis-aligned region $R$ in which a Cox model is well-specified, and there is a similar effect size gap on the hazard function in and outside of $R$, then DDGroup will recover $R$ given a large enough sample. The counterexample in Figure 1 means that this region $R$ may not necessarily have the minimum EPE out of all candidate regions in the dataset, but nevertheless DDGroup will have found a large region with low EPE in which a Cox model fits the data.

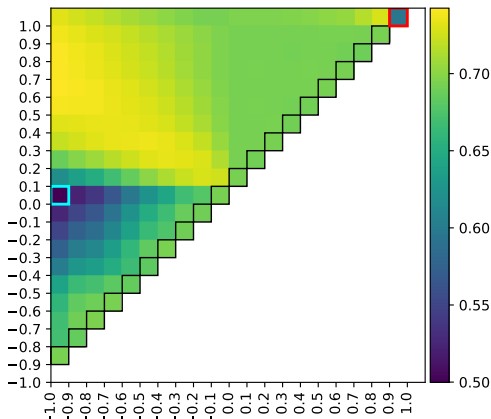

*Figure 2.* Counterexample showing the need for Assumption A4. The cell with coordinates $(a, b)$ at its bottom-left corner denotes the region $[a, b]$. The subgroup $[-1, 0]$ minimizes the EPE and follows a Cox model (cyan boxed cell), but among the regions considered for the core group (black boxed cells), the interval $[0.9, 1]$ has the minimum EPE (red boxed cell).

### G.2. Behavior of the Core Group Selection Procedure

Our primary argument for the validity of Assumption A4 is via the empirical results on synthetic and real data. Nevertheless, for the sake of further understanding the behavior of the core group selection procedure, we also analyze a setting with "lower level" conditions on the data distribution where we can provide guarantees on the core group selection step. This is the content of Proposition G.1.

**Proposition G.1.** *Suppose that for all $t, x, x'$, we have*

$$|\log \lambda(t, x) - \log \lambda(t, x')| \leq |\beta^\top (x - x')|. \tag{46}$$

*Furthermore, suppose there exists $R^\circ \supseteq R^*$ and a constant $c < 1$ such that*

$$|\log \lambda(t, x) - \log \lambda(t, x')| \leq c|\beta^\top (x - x')|$$

*whenever at least one of $x, x' \notin R^\circ$.*

*Let $R_{\text{cs}} \subseteq R^*$ be an axis-aligned box which is a subset of the ground truth region. (This box defines the core group shape, hence the subscript "cs": core shape. For instance, $R_{\text{cs}}$ may be the $\ell_\infty$ ball of radius $\varepsilon$ for some small $\varepsilon$.) Let $\mathcal{R} = \{x_0 + R_{\text{cs}} : x_0 \in \mathbb{R}^d, x_0 + R_{\text{cs}} \subseteq B\}$ be the set of all translations of $R_{\text{cs}}$ which are still contained in the support of the features, i.e., the set of possible core group regions. Then any minimizer of the EPE over $\mathcal{R}$ must be a subset of $R^\circ$, i.e.,*

$$R \in \operatorname*{argmin}_{R' \in \mathcal{R}} \text{EPE}(R') \implies R \subseteq R^\circ.$$

Before we give the proof, we note that the assumptions of Proposition G.1 are incompatible with the assumptions for Theorem 6.1. In particular, it is easy to show that when $\lambda(t, x) = \lambda_0(t)e^{h(x)}$ and $|\lambda(t, x) - \lambda(t, x')| \leq |\beta^\top (x - x')|$ for all $x, x'$, then $h(x)$ must be continuous in $x$. However, the assumptions of Theorem 6.1 (specifically the combination of Assumption A3 and the requirement that $|h(x) - \beta^\top x| \geq C_\varepsilon > 0$ for $x \notin R^*$) imply a discontinuity in $h(x)$ at the boundary of $R^*$. Moreover, Proposition G.1 only implies that the core group is a subset of $R^\circ$ rather than $R^*$. Thus, we emphasize that Proposition G.1 is only meant to provide the reader with additional (theoretical) intuition for the behavior of the core group selection process.

*Proof.* First, note that the EPE for any core group $R = x_0 + R_{\text{cs}} \subseteq R^*$ which is a subset of $R^*$ is equal to the following:

$$\text{EPE}(R) = \mathbb{E}[H(\beta^\top ((X + x_0) - (X' + x_0)))] = \mathbb{E}[H(\beta^\top (X - X'))] := \text{EPE}^*. \tag{47}$$

We can first show that this is indeed the minimum value of the EPE over $R \in \mathcal{R}$. Let $R = x_0 + R_{cs} \in \mathcal{R}$. We have the following:

$$\mathrm{EPE}(R) \geq \mathbb{E}\left[H\left(\log \frac{\lambda(T^*, X + x_0)}{\lambda(T^*, X' + x_0)}\right)\right] \tag{48}$$

$$\geq \mathbb{E}[H(\beta^\top((X + x_0) - (X' + x_0)))] \tag{49}$$

$$= \mathrm{EPE}^*.$$

Inequality (49) holds by combining (46) with the fact that $H(z)$ is decreasing as a function of $|z|$ by Lemma D.1. Thus, core regions which are subsets of the ground truth minimize the expected EPE. It remains to show that if a core region is *not* contained in $R^\circ$, then it must have a strictly larger EPE.

Suppose that $R \not\subseteq R^\circ$. Since $R$ and $R^\circ$ are both closed axis-aligned boxes, it is easy to see that in fact $R \setminus R^\circ$ must have positive Lebesgue measure. Choose $x + x_0 \in R \setminus R^\circ$. The set $S = \{x' : \beta^\top(x - x') = 0\}$ has zero Lebesgue measure, so $(R \setminus R^\circ) \setminus S$ must also have positive Lebesgue measure and is therefore nonempty, so we can choose $x' + x_0 \in (R \setminus R^\circ) \setminus S$.

By Lemma D.1, the entropy is (strictly) decreasing as a function of the absolute value of the logit. But note for $x + x_0$ and $x' + x_0$, and for any value of $t^*$, we have

$$|\log \lambda(t^*, x + x_0) - \log \lambda(t^*, x' + x_0)| \leq c|\beta^\top(x + x_0 - (x' + x_0))| < |\beta^\top(x - x')|,$$

where the final strict inequality holds because $\beta^\top(x - x') \neq 0$ and $c < 1$. Thus, in a neighborhood of $X = x$ and $X' = x'$, we have that

$$H\left(\log \frac{\lambda(T^*, X + x_0)}{\lambda(T^*, X' + x_0)}\right) > H(\beta^\top(X - X')).$$

In particular, with positive probability, the integrand in the lower bound (48) for the EPE of $R$ is strictly larger than the integrand in the expression (47) for the minimum EPE, and it is bounded below by the integrand of $\mathrm{EPE}^*$ (though not necessarily strictly) by the logic of inequality (49). It follows that $\mathrm{EPE}(R) > \mathrm{EPE}^*$, completing the proof. $\square$

# H. Experiment Details

In this section, we give additional details for reproducing the experiments, including complete algorithm implementations, definitions of additional metrics, dataset descriptions, and a brief discussion of compute resources.

## H.1. Algorithms

In this subsection we provide Python-esque pseudocode for the practical implementation of each algorithm. Each method takes the following arguments:

- $X_{\mathrm{adjust}}$: An $n \times d_1$ matrix containing the features to be used by the Cox model.

- $X_{\mathrm{subgp}}$: An $n \times d_2$ matrix containing the features used to define the subgroup. Note that $X_{\mathrm{adjust}}$ and $X_{\mathrm{subgp}}$ can have the same features, partial overlap, or be completely disjoint. In our real data experiments, the features were generally disjoint. In the synthetic experiments, all features were used by both the Cox model and the subgroup definition.

- $Y$: A list of $n$ (event time, failure indicator) tuples, i.e., the $i$-th entry is $(t_i, \delta_i)$.

- $B_{\mathrm{subgp}}$: The bounding box for the subgroup features. In particular, all of the data have $x_{\mathrm{subgp}} \in B_{\mathrm{subgp}}$, and the region $R$ returned by any method must have $R \subseteq B_{\mathrm{subgp}}$.

Each method also accepts algorithm-specific hyperparameters, e.g., the core group size and rejection score quantile for DDGroup, or the maximum tree depth and minimum leaf size for the Cox tree. Given these inputs and valid hyperparameters, each method returns a region $R$ specifying the subgroup. The associated Cox model is always defined by fitting to all training data in $R$.

Some of the algorithms rely on subroutines for which we provide only a "plain English" description, rather than pseudocode. We refer readers to the associated code repository for the full implementations of all of the algorithms.

### H.1.1. BASE

```python
def base(X_adjust, X_subgp, Y, B_subgp):
    return B_subgp
```

*Hyperparameters:* None.

### H.1.2. RANDOM

```python
def random(X_adjust, X_subgp, Y, B_subgp):
    n, d = X_subgp.shape

    selected_points = rng.sample(range(n), 2 * d) # Sample 2d points from X_adjust without
                                                    replacement

    R_subgp = bounding_box(X_subgp[selected_points])

    return R
```

*Hyperparameters:* Random seed for selecting the $2d$ points $\in \{0, \ldots, 99\}$.

### H.1.3. SURVIVAL TREE

```python
def survival_tree(X_adjust, X_subgp, Y, B_subgp, max_depth, min_samples_leaf):
    st = Tree(
        max_depth=max_depth,
        min_samples_leaf=min_samples_leaf,
        splitting_criterion=logrank_statistic
    )
    st.fit(X_adjust, X_subgp, Y) # Standard tree fitting procedure. Recursively choose
                                    splits of X_subgp features which
                                    minimizes the node impurity as measured
                                    by the logrank statistic.

    regions, betas = tree_to_bounding_boxes(st, depth, B) # Returns the bounding boxes and
                                                            Cox models associated to each leaf of
                                                            the tree.

    best_R, best_beta = argmin([EPE(X_adjust, X_subgp, R, beta) for R, beta in zip(regions
                                , betas)]) # Compute the training EPE in
                                            each leaf and select the region with the
                                            lowest EPE.
    return best_R
```

*Hyperparameters:* Maximum tree depth $\in \{1, 2, \ldots, 25\}$, minimum leaf size $\in \{5, 10, 20, 40\}$.

### H.1.4. OPTIMAL SPARSE SURVIVAL TREES

This method uses the implementation of OSST from Zhang et al. (2024). Their code can be found at https://github.com/ruizhang1996/optimal-sparse-survival-trees-public.

```python
def survival_tree(X_adjust, X_subgp, Y, B_subgp, lambd):
    st = OptimalSparseSurvivalTree(regularization=lambd)
    st.fit(X_adjust, X_subgp, Y) # Fitting procedure from Zhang et al.

    regions, betas = tree_to_bounding_boxes(st, depth, B) # Returns the bounding boxes and
                                                            Cox models associated to each leaf of
                                                            the tree.

    best_R, best_beta = argmin([EPE(X_adjust, X_subgp, R, beta) for R, beta in zip(regions
                                , betas)]) # Compute the training EPE in
                                            each leaf and select the region with the
                                            lowest EPE.
```

```
    return best_R
```

*Hyperparameters:* Sparsity regularization strength $\lambda \in \{0.01, 0.02, \ldots, 1\}$.

### H.1.5. COX TREE

```python
def cox_tree(X_adjust, X_subgp, Y, B_subgp, max_depth, min_samples_leaf):
    ct = Tree(
        max_depth=max_depth,
        min_samples_leaf=min_samples_leaf,
        splitting_criterion=EPE
    )
    ct.fit(X_adjust, X_subgp, Y) # Standard tree fitting procedure. Recursively choose
                                       splits of X_subgp features which
                                       minimizes the node impurity as measured
                                       by the EPE.

    regions, betas = tree_to_bounding_boxes(ct, depth, B) # Returns the bounding boxes and
                                                Cox models associated to each leaf of
                                                the tree.

    best_R, best_beta = argmin([EPE(X_adjust, X_subgp, R, beta) for R, beta in zip(regions
                                                , betas)]) # Compute the training EPE in
                                                each leaf and select the region with the
                                                lowest EPE.
    return best_R
```

*Hyperparameters:* Maximum tree depth $\in \{1, 2, \ldots, 25\}$, minimum leaf size $\in \{5, 10, 20, 40\}$.

### H.1.6. PRIM

Refer to (Friedman & Fisher, 1999) for the original algorithm. PRIM relies on two subroutines: peeling and pasting. These can be understood as follows:

**Peeling**

- Start with the full feature space (a large "box").

- Iteratively remove a small fraction $\alpha$ (e.g., 5%) of the data from one face of the box. Each possible peel corresponds to shrinking the box along one variable by trimming off the lowest or highest values.

- Choose the peel that maximally reduces the EPE within the remaining box.

This procedure is iterated until there are no peels which reduce the EPE, or until a minimum size is reached.

**Pasting**

- Start from the bounding box obtained at the end of the peeling procedure.

- Try *adding* a small fraction $\alpha$ of the original data back to one face of the box. Each possible paste corresponds to expanding the box along one variable by including an $\alpha$ fraction of the data which is closest to that face of the box.

- Choose the pasting operation which causes the greatest reduction in EPE. If no pasting operation reduces the EPE, terminate the algorithm and return the current region.

```python
def prim(X_adjust, X_subgp, Y, B_subgp, alpha, min_support_size):
    R = B_subgp.copy()
    current_metric_val = inf
    support = min_support_size * len(X_adjust) # min_support_size given as a fraction of
                                                the total dataset.
```

```python
    # Peeling steps
    while len(in_region(X_subgp, R)) >= support:
        new_R, new_metric_val = peel(X_adjust, X_subgp, Y, R, alpha)

        if new_metric_val >= metric_val:
            break
        else:
            metric_val = new_metric_val
            R = new_R

    R = bounding_box(current_X_subgp)

    # Pasting steps
    while True:
        new_R, new_metric_val = paste(X_adjust, X_subgp, Y, R, alpha)
        if new_metric_val >= metric_val:
            break
        else:
            R = new_R
            metric_val = new_metric_val

    return R
```

*Hyperparameters:* Peeling/pasting parameter $\alpha \in \{0.01, 0.02, \ldots, 0.25\}$, minimum support size $\beta_0 \in \{0.005, 0.01, 0.02, 0.04\}$.

### H.1.7. DDGROUP

*Hyperparameters for DG:* Core group size $\in \{0.05n, 0.1n\}$ ($n$ = size of the training set), rejection threshold quantile $\alpha \in \{0.01, 0.02, \ldots, 0.5\}$.

For the core group selection step, the groups of nearest neighbors are calculated with respect to the Euclidean distance between the features in $X_{\text{subgp}}$.

For the rejection threshold $\alpha$, rather than setting it equal to some absolute constant, we first compute the rejection score for each point in the training data (according to the CRS, C-index, or partial likelihood, depending on the method). Then, points in the bottom $\alpha$-quantile of this collection of scores are rejected; in particular, this means that exactly an $\alpha$-fraction of the points are rejected. We found this to be significantly easier to tune than using an absolute cutoff. We also used this approach for the ablations described below.

**C-Index DDGroup (DG-CI):** DG-CI uses the C-index to measure core group quality, i.e., the neighborhood with the highest C-index is selected as the core group. The C-index is also used to define conformity scores of test points to the core group during the point rejection phase of DDGroup. Specifically, given a core group $\{(x_i, t_i, \delta_i)\}_{i=1}^k$, fit Cox coefficients $\beta$, and a test point $(x^*, t^*, \delta^*)$, we define $y_i = \mathbb{1}\{\beta^\top x_i > \beta^\top x^*\}$ for $i = 1, \ldots, k$. (Note that $1 - y_i = \mathbb{1}\{\beta^\top x_i \leq \beta^\top x^*\}$.) Then the conformity score is

$$s_{\text{CI}}^* = \frac{\sum_{i \,:\, t_i < t^*, \, \delta_i = 1} y_i + \delta^* \sum_{i \,:\, t_i \geq t^*} (1 - y_i)}{|\{i \,:\, t_i < t^*, \, \delta_i = 1\}| + \delta^* |\{i \,:\, t_i \geq t^*\}|},$$

i.e., the fraction of core group points which are concordant with the test point and the given model. We reject points whose score $s_{\text{CI}}^*$ falls below the $\alpha$ quantile of all scores computed over the dataset. *Hyperparameters:* Core group size $\in \{0.05n, 0.1n\}$, rejection threshold quantile $\alpha \in \{0.01, 0.02, \ldots, 0.5\}$.

**Partial likelihood DDGroup (DG-PL):** For this algorithm, we use the partial likelihood to implement the core group and rejection components of DDGroup. Specifically, the core group is selected as the neighborhood with the largest partial likelihood after fitting the Cox model. Given a core group $\{(x_i, t_i, \delta_i)\}_{i=1}^k$, fit Cox coefficients $\beta$, and a test point $(x^*, t^*, \delta^*)$, we define the conformity score

$$s_{\text{PL}}^* = \frac{\exp(\beta^\top x^*)}{\exp(\beta^\top x^*) + \sum_{i \,:\, t_i \geq t^*} \exp(\beta^\top x_i)},$$

when $\delta^* = 1$. Otherwise, if $\delta^* = 0$, we define

$$s^*_{\mathrm{PL}} = \sum_{i \,:\, t_i \geq t^*, \delta_i = 1} \frac{\exp(\beta^\top x^*)}{\exp(\beta^\top x^*) + \sum_{j \,:\, t_j \geq t_i} \exp(\beta^\top x_j)}.$$

That is, the rejection score is the partial likelihood term for the test point when the test point is uncensored; otherwise, it is the sum of all possible partial likelihood terms which are consistent with a censored test observation. As with the other DDGroup versions, we reject points whose score $s^*_{\mathrm{PL}}$ is in the bottom $\alpha$ quantile of scores over the whole dataset. We also briefly remark that the double summation in the $\delta^* = 0$ case can lead to an $\Omega(k^2)$ computation when computed naively. We give an efficient implementation based on sorting the event times $y_i$ and using running partial sums which reduces the cost to $O(k \log k)$ when the times are unsorted, or $O(k)$ when the times are pre-sorted. *Hyperparameters:* Core group size $\in \{0.05n, 0.1n\}$, rejection threshold quantile $\alpha \in \{0.01, 0.02, \dots, 0.5\}$.

Refer to (Izzo et al., 2023) for the DDGroup algorithm in a linear regression context. In our survival analysis setting, we modify the algorithm by using the EPE for the core group selection step and the CRS for the rejection step (or the C-index/partial likelihood alternatives). A high-level overview of the algorithm can be found in the Pythonic pseudocode below.

```python
def ddgroup(X_adjust, X_subgp, Y, B_subgp, core_size, rejection_threshold, core_metric,
                                        rejection_metric):
    core_ind = core_group(X_adjust, X_subgp, Y, core_size, core_metric) # Computes
                                        core_metric on the core_size nearest
                                        neighbors of each point in the dataset.
                                        Nearest neighbors are determined by
                                        Euclidean distance on X_adjust features.
                                        Returns the indices of the neighborhood
                                        with lowest value of core_metric.

    beta_hat = fit_cox(X_adjust[core_ind], Y[core_ind])

    scores = get_scores(X_adjust, Y, beta_hat, core_ind, rejection_metric) # Compute the
                                        rejection metric for each point in the
                                        dataset using the core group selected
                                        above.

    abs_threshold = quantile(scores, rejection_threshold)
    labels = scores < abs_threshold # Reject all points which are in the bottom
                                        rejection_threshold quantile of rejection
                                         scores.

    R = grow_region(X_subgp, labels, B_subgp, center=mean(X_subgp[core_ind], axis=0)) #
                                        Starting from the mean of the core group,
                                         expand the size of the bounding box
                                        until they collide with a rejected point.

    return R
```

DG is implemented with the EPE as the core metric and the CRS as the rejection metric. DG-CI is implemented with the C-index as the core metric and the C-index based rejection scores defined in Section 5. DG-PL is implemented with the partial likelihood as the core metric and the partial likelihood-based rejections scores also defined in Section 5.

### H.1.8. WHY NOT INCLUDE THE CHANGE-PLANE COX MODEL?

The change-plane Cox model, introduced by (Wei & Kosorok, 2018), nominally considers the same problem as our work: subgroup discovery with the Cox model. However, our problem setting differs from theirs in two critical ways.

The first is the manner in which the subgroups themselves are defined. (Wei & Kosorok, 2018) defines two subgroups which are separated by a hyperplane. On the other hand, our work considers subgroups defined as axis-aligned boxes. This approach has precedent in previous works (Izzo et al., 2023). While both the hyperplane and box subgroups are reasonably interpretable, the box subgroups have the added advantage of being standard practice for specifying inclusion criteria for clinical trials, which is an important motivating use case for subgroup discovery. See (Friedman et al., 2015) pg. 129 on

defining patient strata (subgroups); this is precisely the axis-aligned box setting in our framework. Thus, from a purely algorithmic view, our setting and (Wei & Kosorok, 2018) are incomparable—neither subsumes the other—but from a practical perspective, we believe the axis-aligned approach to be superior.

Another important difference between our work and (Wei & Kosorok, 2018) is the modeling assumptions on the data. In particular, (Wei & Kosorok, 2018) assumes a very strict relationship on the two subgroups: specifically, it is assumed that the hazards of the two discovered groups are proportional. In contrast, we make minimal assumptions outside of the region to be discovered other than that the Cox model does not fit it well. Due to the weaker assumptions, this should make our problem definition more challenging but also more practically relevant.

Regarding adapting (Wei & Kosorok, 2018) method to our setting, we believe their proposed algorithm is fundamentally incompatible with our problem. There are two major issues. The first is that their algorithm critically hinges on constructing a "sieve" of potential change planes which define the possible subgroups. The vectors in this sieve consist of normalized eigenvectors of various conditional feature matrices; these eigenvectors will not be axis-aligned in general. Thus, a subgroup defined by the two sides of a plane defined by these vectors will not be an axis-aligned box, making it incompatible with our goal. Second, even if we ignore this difficulty, the way that the final subgroup selection is performed also relies critically on the strong modeling assumption that the two subgroups have hazards which are proportional to each other. As this assumption does not hold in our settings, the selection procedure becomes invalid. Given these many significant obstacles to employing the algorithm for our problem, we excluded it from the suite of algorithms tested.

## H.2. Additional Metric: Rejection Fraction

Let $\{(x_i, y_i, \delta_i)\}_{i=1}^m$ be a subgroup of points with corresponding Cox model $\beta$. For each point $x_i$, we compute the CRS $\tau_i^*$ for $x_i$ with respect to the rest of the points in the subgroup, as defined in Section 4. The *rejection fraction at level $\alpha$* is then

$$\frac{1}{m} \sum_{i=1}^m \mathbb{1}\{\tau_i^* < \alpha\},$$

i.e., the fraction of points in the group whose CRS value is below $\alpha$. A higher rejection fraction indicates that many of the survival time ranks are taking a more extreme value than what is predicted by the model, indicating a poor fit.

## H.3. Datasets

**Table 1** The data were generated according to the counterexample discussed in Section 4.1. Namely, the features are 1D and drawn uniformly from $[0, 1]$. The ground truth hazard function is

$$\lambda(t, x) = \begin{cases} e^{mx} & 0 \leq x < c \\ e^{mx-b} & c \leq x \leq 1 \end{cases}$$

with $m = 10$, $b = 2$, and $c = 0.4$. We generated $n = 4000$ datapoints for the training data.

**Table 2** This dataset used $n = 4000$ datapoints, 2000 used for training and 2000 used for testing, also in dimension $d = 2$. The ground truth region was $R^* = [-1/6^{\frac{1}{d}}, 1/6^{\frac{1}{d}}]^d$ and the ground truth Cox coefficients were $\beta^* = 10 \cdot \mathbf{1}$, where $\mathbf{1} \in \mathbb{R}^d$ is the all-1s vector. Given a feature vector $x \notin R^*$, the nonlinear transformation we used to generate the data outside of $R^*$ was

$$\tilde{x}[1] = 10 \cdot \sin(100 \cdot x[1]^2), \quad \tilde{x}[2] = x[2],$$

where $\tilde{x}[i]$, $x[i]$ denote the $i$-th entries of the transformed features $\tilde{x}$ and the original features $x$, respectively. The survival times outside of $R^*$ were then generated from a Cox model with baseline hazard $\lambda_0(t) \equiv 1$ and Cox coefficients $\beta^{\text{out}} = 0.5 \cdot \mathbf{1}$, but applied to the transformed features $\tilde{x}$.

The real datasets are provided by the sksurv Python package (Pölsterl, 2020).

**AIDS Clinical Trial (AIDS)** This dataset has 1151 samples and extremely high censoring; only 96 (8.3%) of patients are uncensored. The endpoint is the onset of AIDS rather than death for this dataset. We studied three features:

- CD4, a measurement of certain types of white blood cells;

*Table 5.* Comparison of methods on Synth-Nonlinear in higher dimension $d = 4$. Parentheses show standard error. DDGroup (DG) obtains the best performance in terms of recovering $R^*$. We kept the same number $n = 2000$ of training data, making this more challenging than the $d = 2$ setting in the main text. DDGroup still obtains the best performance in terms of recovering $R^*$, but DDG-CI has similar results in this case.

| Method | F1 | EPE | C-Index |
|---|---|---|---|
| Base | 0.29 (0.00) | 0.68 (0.00) | 0.59 (0.00) |
| ST | 0.20 (0.02) | 0.67 (0.00) | 0.59 (0.01) |
| OSST | 0.29 (0.00) | 0.68 (0.00) | 0.59 (0.00) |
| PRIM | 0.30 (0.00) | 0.67 (0.00) | 0.59 (0.00) |
| CT | 0.60 (0.09) | 0.39 (0.07) | 0.81 (0.05) |
| Random | 0.66 (0.04) | 0.54 (0.01) | 0.76 (0.01) |
| DDG-PL | 0.72 (0.04) | 0.41 (0.03) | 0.85 (0.02) |
| DDG-NE | 0.74 (0.02) | 0.40 (0.03) | 0.85 (0.01) |
| DDG-CI | 0.88 (0.02) | 0.26 (0.02) | 0.92 (0.01) |
| **DDGroup** | 0.89 (0.02) | 0.26 (0.02) | 0.92 (0.01) |

- Karnofsky performance scale (Karnof), a measure introduced by (Karnofsky et al., 1948) which measures a patient's ability to perform daily activities; and

- Prior ZDV use (prior), the number of months a patient has used the antiretroviral drug zidovudine.

**German Breast Cancer Study Group 2 (GBSG2)**    This dataset has 686 samples of which 299 (43.6%) are uncensored. We studied the tumor size feature (tsize).

**Veterans' Administration Lung Cancer Trial (VLC)**    This dataset consists of 137 samples, of which 128 (93.4%) are uncensored. We studied one feature, the Karnofsky score, which is a scale that measures a patient's ability to function during the progression of a disease (Karnofsky et al., 1948).

**Worchester Heart Attack Study (WHAS)**    This dataset consists of 500 samples, of which 215 (43%) are uncensored. We studied 3 distinct features: length of stay (los), systolic blood pressure (sysbp), and diastolic blood pressure (diasbp).

### H.4. Additional Results: Higher-Dimensional Synthetic Data

We extended the synthetic data experiment to d=4 dimensions; here the Cox model itself and the subgroup definition are with respect to 4 covariates. We used the same number of datapoints ($n = 2000$) as in the $d = 2$ case, making this a more challenging setup. DDGroup is still able to obtain good results with an F1 score of 0.89 (slightly favoring precision over recall). DDG-CI also works well in this setting.

### H.5. Additional Results: Real Data

In this section, we report the full collection of results on real data. That is, we report the combined results from Table 3 as well as results on the additional real datasets excluded from the main text due to space constraints.

### H.6. NASA Jet Engine Case Study

We performed an additional in-depth case study on the simulated jet engine failure data from (Saxena et al., 2008) to show the utility of the interpretable subgroups discovered by our methods. The data are available through Kaggle.

The goal of the dataset is to predict the remaining useful life (RUL) of a jet engine as a function of different sensor measurements and operating conditions. The data are originally presented as a time series with different sensor measurements recorded as the engine is used. We converted the data into a standard survival analysis format by using each collection of sensor readings as a fixed covariate vector, then using the remaining time until failure from that reading as the survival time. In this dataset, all failures are observed, so there is no censoring.

As discussed by (Saxena et al., 2008), in this dataset, there are known operating conditions (related to different altitudes, flight speeds, and air temperatures) where the relationship between sensor readings and RUL will differ qualitatively. Thus,

*Table 6.* Expanded real results.

| | | Base | Rand | PRIM | ST | OSST | CT | DG-PL | DG-CI | DG |
|---|---|---|---|---|---|---|---|---|---|---|
| AIDS-CD4 | EPE ($\downarrow$) | 0.54 (0.02) | 0.59 (0.13) | 0.53 (0.02) | 0.58 (0.10) | 0.54(0.02) | 0.50 (0.08) | 0.54 (0.09) | 0.59 (0.06) | 0.43 (0.04) |
| | Rej@10% ($\downarrow$) | 0.07 (0.01) | 0.11 (0.05) | 0.06 (0.01) | 0.08 (0.03) | 0.07 (0.01) | 0.12 (0.05) | 0.08 (0.03) | 0.10 (0.02) | 0.07 (0.05) |
| | C-Index ($\uparrow$) | 0.71 (0.01) | 0.72 (0.06) | 0.73 (0.01) | 0.71 (0.04) | 0.71 (0.01) | 0.74 (0.04) | 0.72 (0.05) | 0.68 (0.03) | 0.76 (0.03) |
| | Size ($\uparrow$) | 1.00 (0.00) | 0.17 (0.03) | 0.87 (0.04) | 0.32 (0.11) | 1.00 (0.00) | 0.17 (0.02) | 0.73 (0.14) | 0.55 (0.15) | 0.20 (0.03) |
| AIDS-Karnof | EPE ($\downarrow$) | 0.62 (0.03) | 0.58 (0.11) | 0.62 (0.03) | 0.65 (0.07) | 0.62 (0.03) | 0.72 (0.22) | 0.62 (0.04) | 0.72 (0.22) | 0.38 (0.07) |
| | Rej@10% ($\downarrow$) | 0.09 (0.01) | 0.01 (0.01) | 0.15 (0.01) | 0.11 (0.03) | 0.15 (0.01) | 0.19 (0.02) | 0.07 (0.01) | 0.06 (0.01) | 0.01 (0.01) |
| | C-Index ($\uparrow$) | 0.66 (0.03) | 0.68 (0.07) | 0.66 (0.03) | 0.64 (0.05) | 0.66 (0.03) | 0.69 (0.07) | 0.66 (0.03) | 0.69 (0.07) | 0.84 (0.05) |
| | Size ($\uparrow$) | 1.00 (0.00) | 0.13 (0.01) | 0.96 (0.02) | 0.21 (0.09) | 1.00 (0.00) | 0.44 (0.10) | 0.82 (0.12) | 0.65 (0.14) | 0.15 (0.01) |
| AIDS-prior | EPE ($\downarrow$) | 0.70 (0.00) | 0.67 (0.04) | 0.70 (0.00) | 0.69 (0.05) | 0.70 (0.00) | 0.68 (0.04) | 0.66 (0.03) | 0.70 (0.00) | 0.65 (0.04) |
| | Rej@10% ($\downarrow$) | 0.14 (0.00) | 0.21 (0.05) | 0.06 (0.00) | 0.06 (0.00) | 0.06 (0.00) | 0.06 (0.00) | 0.11 (0.03) | 0.14 (0.00) | 0.13 (0.03) |
| | C-Index ($\uparrow$) | 0.46 (0.02) | 0.55 (0.04) | 0.46 (0.02) | 0.56 (0.04) | 0.46 (0.02) | 0.57 (0.04) | 0.52 (0.06) | 0.46 (0.02) | 0.58 (0.06) |
| | Size ($\uparrow$) | 1.00 (0.00) | 0.16 (0.02) | 0.94 (0.03) | 0.10 (0.01) | 1.00 (0.00) | 0.17 (0.01) | 0.64 (0.15) | 1.00 (0.00) | 0.16 (0.01) |
| GBSG2-tsize | EPE ($\downarrow$) | 0.68 (0.00) | 0.62 (0.04) | 0.68 (0.00) | 0.68 (0.02) | 0.68 (0.00) | 0.63 (0.04) | 0.68 (0.00) | 0.65 (0.03) | 0.61 (0.04) |
| | Rej@10% ($\downarrow$) | 0.14 (0.00) | 0.13 (0.03) | 0.08 (0.01) | 0.04 (0.01) | 0.08 (0.01) | 0.05 (0.02) | 0.13 (0.02) | 0.11 (0.02) | 0.07 (0.02) |
| | C-Index ($\uparrow$) | 0.57 (0.01) | 0.62 (0.04) | 0.55 (0.01) | 0.57 (0.03) | 0.57 (0.01) | 0.62 (0.04) | 0.58 (0.01) | 0.64 (0.05) | 0.64 (0.04) |
| | Size ($\uparrow$) | 1.00 (0.00) | 0.15 (0.02) | 0.94 (0.01) | 0.26 (0.07) | 1.00 (0.00) | 0.11 (0.01) | 0.90 (0.10) | 0.39 (0.14) | 0.12 (0.01) |
| VLC-Karnof | EPE ($\downarrow$) | 0.57 (0.02) | 0.42 (0.14) | 0.58 (0.02) | 0.45 (0.10) | 0.57 (0.02) | 0.22 (0.05) | 0.32 (0.02) | 0.34 (0.06) | 0.33 (0.06) |
| | Rej@10% ($\downarrow$) | 0.04 (0.01) | 0.12 (0.04) | 0.03 (0.01) | 0.07 (0.03) | 0.04 (0.01) | 0.21 (0.04) | 0.11 (0.05) | 0.10 (0.03) | 0.09 (0.03) |
| | C-Index ($\uparrow$) | 0.69 (0.02) | 0.84 (0.06) | 0.68 (0.02) | 0.77 (0.07) | 0.70 (0.02) | 0.93 (0.02) | 0.92 (0.03) | 0.87 (0.03) | 0.87 (0.04) |
| | Size ($\uparrow$) | 1.00 (0.00) | 0.17 (0.02) | 0.88 (0.05) | 0.23 (0.05) | 1.00 (0.00) | 0.20 (0.02) | 0.23 (0.03) | 0.22 (0.02) | 0.28 (0.08) |
| WHAS-DBP | EPE ($\downarrow$) | 0.67 (0.01) | 0.83 (0.07) | 0.67 (0.01) | 0.65 (0.06) | 0.68 (0.01) | 0.64 (0.07) | 0.67 (0.03) | 0.55 (0.08) | 0.62 (0.07) |
| | Rej@10% ($\downarrow$) | 0.07 (0.01) | 0.01 (0.01) | 0.13 (0.00) | 0.21 (0.03) | 0.23 (0.02) | 0.16 (0.03) | 0.04 (0.01) | 0.02 (0.01) | 0.01 (0.01) |
| | C-Index ($\uparrow$) | 0.61 (0.01) | 0.51 (0.06) | 0.61 (0.01) | 0.68 (0.04) | 0.60 (0.02) | 0.66 (0.06) | 0.64 (0.03) | 0.75 (0.07) | 0.70 (0.06) |
| | Size ($\uparrow$) | 1.00 (0.00) | 0.14 (0.01) | 0.98 (0.01) | 0.15 (0.02) | 0.64 (0.08) | 0.16 (0.01) | 0.62 (0.15) | 0.20 (0.09) | 0.12 (0.01) |

the different operating conditions serve as natural subgroups of the data which we can try to recover. Note that the data are created by simulating physical processes leading to the fault of the engine; in particular, the simulation has nothing to do with the Cox model at face value, thereby testing the robustness of our methods to misspecification.

To create the dataset, we subsampled data where the flight speed and air temperature were in a fixed state, with the goal to recover the two possible altitude operating conditions. The creation of a new datapoint from every time point in the original time series leads to a very large dataset. Thus, from among the remaining data, we further randomly subsampled down to $n = 5000$ datapoints to avoid excessive runtimes. As in the other experiments, we perform an 80-20 train/test split and average all results over 10 such splits.

The Cox model adjusts for two sensor readings: the engine core speed ("Nc," in RPM) and the corrected fan speed ("NRf," also in RPM).[2] Since the data were not uniformly distributed over the covariate space, we measure the precision and recall of a discovered subgroup according to the number of datapoints in the ground truth and estimated regions, rather than the region volumes. Specifically, given a ground truth region $R^*$, an estimate $\hat{R}$, and the dataset of features $\{x_i\}_{i=1}^n$, we define

$$\text{Precision}(\hat{R}, R^*) = \frac{\#\{i \in [n] \,:\, x_i \in \hat{R} \cap R^*\}}{\#\{i \in [n] \,:\, x_i \in \hat{R}\}},$$

$$\text{Recall}(\hat{R}, R^*) = \frac{\#\{i \in [n] \,:\, x_i \in \hat{R} \cap R^*\}}{\#\{i \in [n] \,:\, x_i \in R^*\}}.$$

In the dataset, there are two potential ground truth regions $R^*_{\text{lo}}$ and $R^*_{\text{hi}}$ corresponding to lower and higher altitude flights. Given an estimate $\hat{R}$, to compute the precision, we compute $\text{Precision}(\hat{R}, R^*_{\text{lo}})$ and $\text{Precision}(\hat{R}, R^*_{\text{hi}})$ and report the max. The same procedure is repeated for the recall. In principle, this means that the precision and recall could be computed against two different ground truth subgroups. However, when the precision is 1, this means that $\hat{R}$ was a subset of either $R^*_{\text{lo}}$ or $R^*_{\text{hi}}$. Since these two ground truth regions are disjoint, the recall will also be computed against the same ground truth region.

The results are shown in Table 4. Several of the methods find subgroups with perfect precision. This means that the subgroup belongs entirely to one of the two possible operating conditions, which is ideal; there is complete separation between the two

---

[2]Note: In modern jet engines, the core and fan are two separate components which can rotate at different speeds, leading to potentially different values for these covariates. See https://www.grc.nasa.gov/WWW/K-12/airplane/aturbf.html for an explanation.

qualitatively different regimes. The Cox tree performs best on this dataset, obtaining a low EPE value and the highest recall among methods with perfect precision. Some of the baseline methods (Random and the C-index ablation of DDGroup) find subgroups with slightly lower EPE than DDGroup, at the cost of identifying a smaller fraction of the ground truth region (lower recall). We remark that because the Cox model is not necessarily a perfect fit to this data (which was generated via simulations of physical equations), we do not expect that all of the data within a single operating regime will be grouped together by the Cox model.

The methods which obtain perfect precision all return subsets of the high altitude operating conditions. Interestingly, there is a *qualitative* as well as quantitative change in the Cox coefficients for these two settings. Without accounting for the subgroup, both coefficients have small positive values. However, for the high altitude subgroups, the coefficient on the core speed has a larger positive value, while the coefficient on fan speed becomes negative. There is a natural physical interpretation of this observation. On average, increased usage of both the fan and core components will lead to degradation of the engine due to mechanical wear; this is reflected by the small positive coefficients of the Cox model fit to the whole dataset. However, by taking a subgroup approach, we can consider a *different baseline hazard function* for different subgroups. High altitude will likely require greater engine utilization to maintain overall. However, conditional at flying at high altitude, a greater usage of the fans may lead to less strain on the engine than relying on the core. Thus, the subgroup analysis approach suggests an actionable plan for engine usage at high altitudes: increase reliance on the fan, and reduce reliance on the core if possible. This indeed seems to be the trend in modern jet engines which have a reduced core size (see, e.g., https://www.nasa.gov/aeronautics/smaller-is-better-for-jet-engines/). We remark that as with any data-driven approach to hypothesis generation, these conclusions should be verified with domain experts and checked experimentally.

### H.7. Compute Resources

Our experiments consisted of a large number of lightweight individual jobs–one per combination of (dataset, train/test split, subgroup discovery method, hyperparameter setting). We used an internal SLURM-managed compute cluster and only CPU hardware for these jobs, specifically, AMD EPYC 7542 and 7543 32-Core Processors, Intel(R) Xeon(R) Silver 4108 CPU @ 1.80GHz, and similar. No GPUs were required for these experiments.

