# OpenReview forum: "Subgroup Discovery with the Cox Model"
_ICML.cc/2026/Conference — ICML 2026 regular_

### Official Review · Reviewer_snGA · 2026-02-23

**Soundness:** 3
**Presentation:** 2
**Significance:** 2
**Originality:** 2
**Overall Recommendation:** 3
**Confidence:** 3

**Summary:**

The paper studies subgroup discovery for survival analysis, aiming to identify interpretable (axis-aligned) subgroups in which the Cox model provides a strong fit. Building on the DDGroup framework of Izzo et al. (2023), the authors adapt and formalize Expected Prediction Entropy (EPE) as a quality measure for survival models and use Conditional Rank Statistics (CRS) to assess whether individual points are consistent with a candidate subgroup’s Cox model. They analyze theoretical properties of EPE and CRS in the context of subgroup discovery and establish recovery guarantees under strong assumptions. Empirically, they compare their approach to several baselines, including fitting a Cox model on the full data (no subgroup), random subgroups, survival trees, Cox trees, and PRIM. Their method performs best on synthetic datasets, successfully recovering ground-truth subgroups, while results on real-world datasets are more mixed.

**Compliance With Llm Reviewing Policy:**

Affirmed.

**Final Justification:**

The paper studies an interesting problem and proposes methods based on EPE and CRS within this setting. The rebuttal clarifies several points, particularly regarding baselines and experimental design choices.

However, my main concern regarding the scope and practical formulation remains. The approach focuses on identifying a single subgroup (and in the real-data experiments, one defined by a single variable), which I find unnecessarily restrictive. Given the proposed criteria (EPE and CRS), it seems natural to use them to identify multiple regions where the Cox model fits well, for example via recursive partitioning or model-based tree approaches that fit Cox models within the leaves. Importantly, this would still yield local, interpretable models within each subgroup, while at the same time covering the full population and capturing heterogeneity across it. In contrast, restricting the analysis to a single subgroup limits the practical usefulness of the approach.

The rebuttal did not change my assessment on this point, and I therefore maintain my current score (weak reject).

**Key Questions For Authors:**

In addition to the ones stated above:
- In the synthetic experiments, ST and PRIM perform unexpectedly poorly, in some cases worse than the random baseline. Could the authors clarify why this happens? Is this due to the optimization criterion, hyperparameter choices, or properties of the synthetic setup?
- In the real-data experiments, subgroups are defined using only a single pre-selected variable. What is the rationale for this restriction? Would allowing multiple variables for subgroup definition change the results or the interpretability of the approach?

**Limitations:**

The paper does not sufficiently address what I consider its main limitation: the method explicitly focuses on identifying a single subgroup in which the Cox model fits well, while the remainder of the data—where the model may not be appropriate—is not analyzed or discussed. The implications of ignoring these regions, and how they should be interpreted or modeled in practice, are not considered.

**Strengths And Weaknesses:**

Strengths
- The metrics EPE and CRS are clearly defined and well motivated within the subgroup discovery framework. Their respective roles in core group selection and rejection are conceptually coherent.
- The theoretical properties and assumptions are stated transparently, and full proofs are provided in the appendix.
- The experimental setup is clearly described, and the authors evaluate their method against multiple baselines on both synthetic and several real-world datasets.


Major weaknesses:
- The overall modeling objective is not fully convincing. The paper focuses on identifying a single subgroup in which the Cox model provides a strong fit, but it remains unclear how this addresses the broader modeling problem. In particular, the regions where the Cox model does not fit well are effectively ignored. In most practical scenarios, it would be preferable to model the entire population while explicitly capturing heterogeneity. This can be achieved either through flexible ML approaches with post-hoc explanations or through interpretable models such as model-based trees that fit Cox (or other parametric) models within nodes and account for interactions and nonlinearities via splits. The paper does not sufficiently position its contribution relative to these alternatives.
- The background section is too brief for a paper of this technical nature. Key concepts from survival analysis and, in particular, the DDGroup framework are not adequately introduced in the main text. Important definitions and explanations are deferred to the appendix, which makes the core contribution harder to understand and weakens the self-contained nature of the paper.
- Although the theoretical assumptions are clearly stated, they are quite strong and often unrealistic in practical settings (e.g., distributional assumptions and well-specified linear structure). While it is acceptable to present theory under simplifying assumptions, the paper lacks a thorough discussion of how violations of these assumptions affect the method in practice. The comparatively mixed results on real datasets suggest that this issue is non-trivial, but this discrepancy is not sufficiently analyzed.

Minor weaknesses:
- The proof of Proposition 3.2 is difficult to follow and would benefit from a more detailed and intuitive explanation.
- The description of the main algorithm in the core paper is relatively high-level. While it is reasonable to place detailed baseline descriptions in the appendix, the proposed method itself should be explained more explicitly in the main text (e.g., clearer pseudocode or a structured step-by-step outline).

---

> ### Author Rebuttal · Authors · 2026-03-25
>
> Thank you for your thoughtful review. We address each of your concerns below.
>
> >Modeling objective is not convincing
>
> We discuss this modeling choice in the introduction. Whether this is appropriate depends on the application. If the primary goal is prediction, then higher-capacity models and post-hoc explainability methods may be preferred. However, we discuss many scenarios where the primary goal is not *only* prediction, but where the discovered subgroups themselves are also valuable.
>
> There is an entire body of literature dedicated to the problem of subgroup discovery; this field is essentially dedicated to the problem of dealing with “interesting” subsets of the data, rather than global models (see Atzmueller, 2015, cited in the paper). Subgroup discovery is especially relevant in clinical settings. The FDA’s guidance on enrichment strategies for clinical trials (https://www.fda.gov/media/121320/download) explicitly recommends identifying patient subgroups most likely to benefit. Our framework directly addresses this use case.
>
> We recognize that the subgroup approach is not mainstream in the ML community, and we do not mean to imply that other approaches should be ignored. Modeling subsets of the data is a complementary alternative approach.
>
> Lastly, some of our methods–namely, the CT and ST baselines–do give global models if this is desired.
>
> >Background section and description of the main algorithm are too brief
>
> We would have liked to include more background material in the main text. The decision to move it to the appendix was a necessity from trying to cram 36 pages of material into the 8 page limit. An algorithm box for DDGroup was sacrificed for the same reason: we used the main text to describe and explore the novel parts (EPE and CRS) in greater detail. If the paper is accepted, we will use the additional page to include more background as well as a full description of DDGroup.
>
> >Strong/unrealistic theoretical assumptions
>
> While the focus of our paper is theoretical, we hope that the real and synthetic data experiments provide a convincing proof-of-concept that our methods have practical applicability. This was the main motivation for the NASA case study: we are able to recover subgroups of the jet engine simulation data which are qualitatively aligned with real design decisions in modern engines, even though we do not expect our theoretical assumptions to hold in this data.
>
> We also mention an important point which is mentioned in the “Results” paragraph at the bottom of pg. 7: DDGroup is on the Pareto frontier of (EPE, rejection fraction) for *every* real dataset, meaning that it is never dominated by another method in terms of both predictive power and plausibility of the Cox model. Since DDGroup is not purely optimizing for EPE, this is the more important feature of DDGroup: good predictive power *and* good fit of the Cox model.
>
> >The proof of Proposition 3.2 is difficult to follow
>
> We have added more explanation to the text, though we do not include it here due to the rebuttal limit.
>
> >ST and PRIM perform poorly on synthetic data
>
> The Random baseline is stronger than it appears: it evaluates 100 randomly sampled regions and selects the one with lowest training EPE, making it a randomly directed search rather than a random model.
>
> For ST, the tree splitting criterion is misaligned with the goal of fitting a Cox model. The ST split is defined to increase the survival heterogeneity between the resulting children nodes (via the log-rank criterion). The CT approach tries to fix this problem by aligning the splitting criterion with the end goal and using the EPE, and we see that this improved alignment generally yields improved results. For PRIM, it has been our experience that this algorithm does not perform very well in a variety of subgroup discovery tasks. It was an early algorithm which is almost entirely heuristic in nature. We included it for completeness, but we generally don’t recommend using it.
>
> >Subgroups are defined using only a single pre-selected variable in the real data experiments
>
> Using multiple variables to define the subgroup boundaries can be done in principle (and we do this in the Synth-nonlinear dataset). The decision to use only single variables for the real data was based on common practice in defining patient subgroups for clinical trials, one of the motivating use cases for our methods. (Refer to Friedman et al., Fundamentals of clinical trials, 2015.) It is preferred that the subgroup descriptions should be as simple as possible: increasing the number of variables used to define the subgroup further restricts the eligible population and makes the subgroup less interpretable.
>
> We specifically chose age as the subgroup variable because it is known to interact with other biomarkers for many health-related outcomes (see e.g. Belloy et al., Mak et al., Moqri et al. cited in the Datasets paragraph of Section 7.2), so we can expect subgroups with respect to age to exist.

---

> > ### Author Rebuttal · Reviewer_snGA · 2026-04-02
> >
> > Thank you for the detailed and thoughtful responses. The rebuttal clarifies some concerns, however, my main concern about the modeling objective remains insufficiently addressed. While subgroup discovery is a valid paradigm, the paper does not discuss how regions outside the identified subgroup—where the Cox model may not fit well—should be interpreted or handled.
> >
> > More fundamentally, the proposed objective could be embedded into a more comprehensive framework (e.g., recursive partitioning or model-based trees) to obtain multiple interpretable regions with well-specified Cox models, thereby covering the full population. In this light, focusing on a single subgroup appears unnecessarily restrictive.
> >
> > Additionally, restricting subgroup definitions to a single pre-selected variable in the real-data experiments seems overly limiting, as subgroup structure in practice is often driven by interactions between multiple variables.
> >
> > Given that this concerns the core positioning and applicability of the method, I maintain my current score.

---

> > > ### Author Response · Authors · 2026-04-02
> > >
> > > Thank you for your detailed response. We address the remaining concerns below.
> > >
> > > ### **On single-subgroup focus vs. full-population coverage:**
> > >
> > > We respectfully note a tension in the review: the reviewer acknowledges that "subgroup discovery is a valid paradigm" but maintains that focusing on a single subgroup is "unnecessarily restrictive." However, partial coverage is an inherent and accepted feature of subgroup discovery, not a limitation of our method specifically. As [Atzmueller et al. (2005)](https://www.ijcai.org/Proceedings/05/Papers/1217.pdf) state: "Subgroup discovery does not necessarily focus on finding complete relations; instead partial relations, i.e., (small) subgroups with 'interesting' characteristics can be sufficient." For another example, see the survey by [Helal (2016)](https://jcst.ict.ac.cn/en/article/pdf/preview/10.1007/s11390-016-1647-1.pdf), which empirically evaluates many subgroup discovery algorithms and explicitly measures coverage as a metric. All algorithms surveyed achieve coverage well below 100%, confirming that partial population coverage is the norm, not the exception, in this field.
> > >
> > > We also note that our contributions can be embedded into methods which fully partition the data. Our EPE and CRS are general-purpose tools, and in fact, our Cox Tree baseline already demonstrates this by using the EPE as a splitting criterion within a tree. The theoretical contributions (proper scoring rule properties of EPE, convergence of CRS) are independent of whether one seeks a single subgroup or a full partition.
> > >
> > > ### **On age-only subgroup definitions:**
> > >
> > > Age-only eligibility criteria are standard practice in clinical research. The FDA has issued dedicated guidance documents for age-based subgroups, including ["Inclusion of Older Adults in Cancer Clinical Trials" (2022)](https://www.fda.gov/regulatory-information/search-fda-guidance-documents/inclusion-older-adults-cancer-clinical-trials) and ["Minimum Age Considerations for Inclusion of Pediatric Patients" (2020)](https://www.fda.gov/regulatory-information/search-fda-guidance-documents/cancer-clinical-trial-eligibility-criteria-minimum-age-considerations-inclusion-pediatric-patients), reflecting the regulatory importance of age as a standalone stratification variable. The [ICH E7 guideline](https://www.fda.gov/regulatory-information/search-fda-guidance-documents/e7-studies-support-special-populations-geriatrics-questions-and-answers) similarly focuses on geriatric subpopulations defined solely by age. These are all single-variable age thresholds, rather than multi-variable interaction-based definitions, exactly as in our experiments. We agree that multi-variable subgroup definitions can be valuable in practice, and our methods support this (as demonstrated in the synthetic experiments). However, the existence of these FDA guidance documents shows that age-only subgroups already have standalone practical utility. Since our real-data experiments are intended as a proof of concept for the framework rather than a definitive clinical analysis, we believe age-based stratification is sufficient for this purpose.
> > >
> > > We hope that we have provided sufficient objective evidence that both the single-subgroup modeling objective and the age-based experimental design have considerable precedent in the subgroup discovery and clinical trials literature. We would also like to reiterate that the core technical contributions of the paper—the EPE, CRS, and their theoretical properties—are applicable regardless of whether one targets a single subgroup or a full partition, and we believe these contributions stand on their own merits.

---

### Official Review · Reviewer_iuQD · 2026-03-13

**Soundness:** 4
**Presentation:** 4
**Significance:** 2
**Originality:** 4
**Overall Recommendation:** 5
**Confidence:** 3

**Summary:**

This paper studies subgroup discovery for survival analysis using the Cox proportional hazards model. It introduces two new metrics—Expected Prediction Entropy (EPE) and Conditional Rank Statistics (CRS)—to identify interpretable regions where the Cox model fits well. The proposed algorithms extend the DDGroup framework and demonstrate improved subgroup recovery and predictive performance on synthetic and real datasets.

**Compliance With Llm Reviewing Policy:**

Affirmed.

**Key Questions For Authors:**

1.	For the real datasets, can the authors provide additional evidence that the discovered subgroups correspond to clinically meaningful populations rather than artifacts of the subgroup discovery process?
2.	In the experiments, subgroups are defined primarily using age. How sensitive are the results to this design choice? Have the authors tested subgroup discovery using other covariates?
3.	How does defining subgroups by age affect the interpretation of other covariates in the Cox model? Could this introduce bias in estimated hazard ratios?

*I should note that in my own work I typically use the Cox model to evaluate survival differences across clusters or discovered subtypes, rather than to directly discover subgroups as done here. Therefore I may not be an expert in Cox-based subgroup discovery. If any of my concerns reflect misunderstandings or naive assumptions, I would appreciate clarification from the authors. I nevertheless found the paper very interesting and enjoyable to read.*

**Limitations:**

The paper acknowledges several limitations. In particular, the authors note that the EPE metric depends on the feature distribution, which may complicate comparisons across subgroups. They also highlight the need for valid statistical inference and p-values for discovered subgroups, which remains an open research problem.

**Strengths And Weaknesses:**

**Strengths**

1. Well written and technically thorough

The paper is clearly written and provides detailed theoretical analysis of both EPE and CRS, including properties such as proper scoring behavior and convergence guarantees.

2. Novel evaluation metric

The introduction of Expected Prediction Entropy (EPE) is a meaningful contribution, addressing known limitations of commonly used metrics such as the C-index and partial likelihood.

3. Interesting statistical construction

The Conditional Rank Statistics (CRS) provide an elegant mechanism for testing whether new data points plausibly follow the same Cox model as a reference subgroup.

4. Empirical performance

Experiments on synthetic and real-world datasets show consistent improvements over baseline approaches, and the synthetic experiments demonstrate that the proposed method can recover the true subgroup under well-specified conditions.

**Weaknesses**

1. Title may be overly broad

While the paper focuses on subgroup discovery using the Cox model, the primary novelty lies in the introduction of EPE and CRS. The current title suggests a broader methodological framework for Cox subgroup discovery, which may somewhat obscure the main contribution.

2. Limited discussion of clinical interpretation

In the real-data experiments, subgroups are primarily defined using age thresholds, while the Cox model is fit using other covariates. The clinical interpretation and validation of these discovered subgroups remain unclear.
In particular, it would be helpful to understand:
•	whether these subgroups correspond to meaningful clinical populations
•	how robust the findings are beyond age-based stratification.

3. Lack of ground truth in real datasets
While synthetic experiments provide ground-truth validation, real datasets rely mainly on proxy metrics (EPE, C-index, rejection fraction). Without external validation or known subgroup structure, it is difficult to assess whether the discovered subgroups correspond to meaningful underlying population heterogeneity. People normally used disease subtype classification as ground truth for testing, for example, PAM50 subtype in breast cancer.

4. Potential confounding from subgroup-defining variables

Since the subgroup is defined using age while other covariates are used in the Cox model, it is unclear how the subgroup selection interacts with the estimation of other model coefficients. This raises questions about how the subgroup definition may affect interpretation of other variables in the model.

---

> ### Author Rebuttal · Authors · 2026-03-25
>
> Thank you for your thoughtful review, we are very glad that you enjoyed the paper! We especially appreciate your candid note about your perspective, and we hope our responses help clarify the Cox subgroup discovery setting.
>
> >Title may be overly broad
>
> While it may unfortunately be too late to change the paper title, we hope that the abstract and introduction adequately emphasize the main novelty (EPE and CRS). We hope that the breadth of methods introduced can also partially account for the breadth of the title as well.
>
> >Limited discussion of clinical interpretation/lack of ground truth in real datasets
>
> We agree that clinical validation of patient subgroups (or general verification by a domain expert in other settings) is a critical aspect of this overall problem. We attempted to give a flavor of this complete process with the NASA case study. We consider the paper’s primary contributions to be theoretical, and as we are already quite space-constrained, we hope the results on clinical data can be accepted more as an additional proof-of-concept rather than fully validated empirical discoveries.
>
> >Sensitivity to age as the subgroup variable
>
> We chose age as the subgroup variable because it is known to modulate/interact with other biomarkers for many health-related outcomes (see e.g. Belloy et al., Mak et al., Moqri et al. cited in the Datasets paragraph of Section 7.2), so we can expect subgroups with respect to age to exist. In early experiments, we tested the methods using other variables to define the subgroups, but we found that in many cases it appeared that subgroups with respect to these other markers might simply not exist, as even exhaustive searches did not find significant improvement over the baseline. This is one of the inherent difficulties of subgroup discovery: without access to a ground truth, it can be very hard to tell if your method is failing or the subgroup simply isn’t present.
>
> >Potential confounding from subgroup-defining variables
>
> We believe that the most reasonable interpretation of the coefficients is simply as a local model. Consider the following: in a “normal” survival analysis setting, when we are collecting the data, there is already an “implicit” subgrouping procedure taking place. For instance, we will only collect data from patients at a hospital or who have some qualifying symptoms, rather than patients from the overall population. When training a Cox model on data collected in this way, we interpret it as a model *for the population that the data were collected from*; the coefficients define the change in risk for members of this population. Our scenario is essentially the same, except (1) the subgrouping is explicit rather than implicit and (2) since the subgroup is defined after observing the data, there may be independence introduced. We agree that the second point is important and merits future study; this is very similar to the “valid p-value” issue we list in the Limitations & Future Work section of the conclusion. For now, we hope that by evaluating the discovered subgroups on held-out test data (which was not seen when the subgroup was defined, and therefore doesn’t suffer from difference (2)) we have partially addressed this issue.
>
> Thank you again for your constructive feedback. If you have any remaining questions, please do not hesitate to ask.

---

> > ### Author Rebuttal · Reviewer_iuQD · 2026-04-03
> >
> > Thanks for the rebuttal, all my concerns have been adequately addressed.

---

### Official Review · Reviewer_N8Aa · 2026-03-16

**Soundness:** 3
**Presentation:** 4
**Significance:** 2
**Originality:** 3
**Overall Recommendation:** 4
**Confidence:** 3

**Summary:**

This paper studies the problem of subgroup discovery for survival analysis, where the goal is to find an interpretable subset of the data on which a Cox model is highly accurate. The authors identify C-index and partial likelihood are not proper metrics for such subgroup discovery problem and proposed expected prediction entropy (EPE) and conditional rank statistics (CRS). The paper proposed a method that uses EPE for the core group selection and the CRS for the group expansion.

**Compliance With Llm Reviewing Policy:**

Affirmed.

**Final Justification:**

My main concerns have been addressed by author's responses, therefore I increased my score. And I do hope authors will add the literatures and experiments i mentioned in the final version.

**Key Questions For Authors:**

- In the expansion phase, is 10% an arbitrary threshold chosen to prevent overfitting?
- Is the metric “size” in table 3 refer to subgroup size?
- What does the result look like if multiple covariates in the original real survival data are used?
- Given that your method needs very strong assumptions, how would you justify it can be useful in practice?

I am willing to increase my score if these questions are addressed.

**Limitations:**

yes

**Strengths And Weaknesses:**

Strengths:
- The authors clearly define the subgroup discovery for suvival analysis problem they try to solve and state a list of assumptions for their method, which is well-specified.
- The authors identified the shortcomings of existing metrics for survival subgroup discovery problems and provided a visualized example for it, which helps reader understand the counter-examples.
- Theoretical analysis and proofs are provided, and most of them are clear to me.
- The new methods perform very well on synthetic data.

Weaknesses:
- There is not enough literature, please include more recent works, for example:
   - Deep learning based:
     - Katzman, Jared L., et al. "DeepSurv: personalized treatment recommender system using a Cox proportional hazards deep neural network." BMC medical research methodology 18.1 (2018): 24.
     - Ching, Travers, Xun Zhu, and Lana X. Garmire. "Cox-nnet: an artificial neural network method for prognosis prediction of high-throughput omics data." PLoS computational biology 14.4 (2018): e1006076.
     - Che, Zhengping, et al. "Recurrent neural networks for multivariate time series with missing values." Scientific reports 8.1 (2018): 6085.
     - B. D. Ripley and R. M. Ripley. Neural networks as statistical methods in survival analysis. Clinical Applications of Artificial Neural Networks, 237:255, 2001
     - E. Giunchiglia, A. Nemchenko, and M. van der Schaar. Rnn-surv: A deep recurrent model for survival analysis. In Artificial Neural Networks and Machine Learning–ICANN 2018: 27th International Conference on Artificial Neural Networks, Rhodes, Greece, October 4-7, 2018 Proceedings, Part III 27, pages 23–32. Springer, 2018

   - Cox model literature:
     - Liu, Jiachang, Rui Zhang, and Cynthia Rudin. "FastSurvival: Hidden computational blessings in training Cox proportional hazards models." Advances in Neural Information Processing Systems 37 (2024): 87712-87765

- I believe the horizontal dashed red line in Fig. 1 should be at 0.4 instead of 0.5.
- There are several more modern survival tree methods can be evaluated as baselines:
   -  Bertsimas, D., Dunn, J., Gibson, E., and Orfanoudaki, A. Optimal survival trees. Machine learning, 111(8):2951–
3023, 2022.
   - Zhang, R., Xin, R., Seltzer, M., & Rudin, C. (2024). Optimal sparse survival trees. Proceedings of machine learning research, 238, 352.
   - Huisman, Tim, Jacobus GM van der Linden, and Emir Demirović. "Optimal survival trees: A dynamic programming approach." Proceedings of the AAAI Conference on Artificial Intelligence. Vol. 38. No. 11. 2024.
- Even though the authors claim their method achieves 2/3 of best EPE in real survival data, but there are in total only 4 datasets are evaluated and their method is not among the best methods on dataset “WHAS” and “VLC”. This claim is somehow misleading as dataset “AIDS” could be biased and their method indeed perform very well on three different versions of “AIDS”.

- minor: In Section 3.2 it can be changed to ” two independent units”,  to let it clear why equation 1 holds.

---

> ### Author Rebuttal · Authors · 2026-03-25
>
> Thank you for your thoughtful review.
>
> >Missing references
>
> Some of these are already included: we cite DeepSurv (Katzman et al.) and optimal survival trees (Bertsimas et al.). We have added the other references.
>
> >Fig. 1
>
> In Fig. 1, the subgroup (interval) [a, b] is represented by the box whose *bottom-left* corner is at coordinate (a, b) in the graph. If the horizontal dashed red line were placed at 0.4, it would exclude the region [0, 0.4], but the boxes below the dashed red line are all of the subsets of [0, 0.4], so this should not be excluded. Instead, we want to start excluding regions with right endpoint 0.5 or larger, so the line should be at y=0.5.
>
> >Missing baselines
>
> The methods listed are based on optimal survival trees. The difference between these methods and the standard survival tree (ST, included as a baseline) is in how the tree is grown. The standard ST uses a greedy criterion at each level of the tree, creating locally optimal improvements in the selected criteria; OSTs may backtrack to find a globally optimal split.
>
> There is a consistent trend in the cited OST papers: replacing the greedy splits with optimal splits does improve performance, but by a small amount (see Fig. 5 of Bertsimas, Fig. 3 of Zhang, and Fig. 3 of Huisman; in fact, in the Huisman paper, there doesn’t appear to be any significant gap between the standard approach and the OST at all). As a result, we can safely conclude that adding the OST approach will not qualitatively change our empirical results. OST can be considered a refinement of our ST or CT methods. ST never obtains the best performance and is usually inferior by a large margin. The performance of CT is bimodal: it either gives the best predictions by a wide margin, or else lags behind other methods. Small optimizations from the tree splits will not change these qualitative outcomes.
>
> We will cite these works and include this discussion in the paper. However, given the complexity of implementing the OST methods; the modest change in performance, which is highly unlikely to change the qualitative conclusions; and the fact that we have already created and implemented 7 methods in what is primarily a theoretical paper; we hope that you can forgive our exclusion of the OST approaches.
>
> >Regarding the claim that DDGroup has the best EPE on 2 / 3 datasets
>
> The “2 / 3 datasets” just refers to the 3 datasets in Table 3 in the main text. We will change the wording to make this clearer.
>
> We would also like to emphasize an important point from the “Results” paragraph at the bottom of pg. 7: DDGroup is on the Pareto frontier of (EPE, rejection fraction) for *every* real dataset, meaning that it is never dominated by another method in terms of both predictive power and plausibility of the Cox model. Since DDGroup is not purely optimizing for EPE, this is the more important feature of DDGroup.
>
> One final note is that, although DDGroup is the primary method that we focus on, *all* of the methods that we compare are novel contributions of the paper, as they all require at minimum a novel combination of components, or entirely novel components such as the EPE and CRS, to be implemented and applied to this problem.
>
> >In the expansion phase, is 10% an arbitrary threshold chosen to prevent overfitting?
>
> Yes, this is correct. Without specifying a minimum size, one can get very low training EPE due to overfitting: if there are fewer than d (= dimension of the Cox model) units with linearly independent features in the training group, then a Cox model fit to these points can achieve arbitrarily low EPE, but this will be completely overfit.
>
> >Is the metric “size” in table 3 refer to subgroup size?
>
> Yes, this is the fraction of the dataset which is contained in the subgroup.
>
> >Results with multiple covariates
>
> The NASA case study gives an example of this: in this case, the Cox model is fit to two-dimensional features. We also have an example where the subgroup is defined with more than one feature on the Synth-nonlinear dataset. The decision to use only single variables for the real data was based on common practice in defining patient subgroups for clinical trials, one of the motivating use cases for our methods. Generally, it is preferred that the subgroup descriptions should be as simple as possible: increasing the “entry requirements” (i.e., number of variables used to define the subgroup) further restricts the eligible population and makes the subgroup less interpretable.
>
> >Strong assumptions/practical utility
>
> While the focus of our paper is theoretical, we hope that the real and synthetic data experiments provide a convincing proof-of-concept that our methods have practical applicability. This was the main motivation for the NASA case study: we are able to recover subgroups of the jet engine simulation data which are qualitatively aligned with real design decisions in modern engines, even though we do not expect our theoretical assumptions to hold in this data.

---

> > ### Author Rebuttal · Reviewer_N8Aa · 2026-04-04
> >
> > Thank you for your response, but I have a few more questions.
> >
> > > given the complexity of implementing the OST methods; the modest change in performance, which is highly unlikely to change the qualitative conclusions
> >
> > All these three OST algorithms provide implementation. I understand modifying their code to run experiments in subgroup discover requires some effort, it is still worth comparing with them though.
> >
> > > DDGroup is on the Pareto frontier of (EPE, rejection fraction) for every real dataset
> >
> > I agree with that, but CT is also on the Pareto frontier for dataset `GBSG2` and `VLC`. I meant to ask what stopped you from evaluating on more real-world survival data to show stronger evidence to support your claim? What's the advantage of DDGroup over CT on those two datasets?
> >
> > > The NASA case study gives an example of this: in this case, the Cox model is fit to two-dimensional features
> >
> > Can DDGroup generalize well (in terms of EPE, C-index, rejection and speed) to datasets with features more than just 2?

---

> > > ### Author Response · Authors · 2026-04-06
> > >
> > > Thank you for your continued engagement with the paper.
> > > ### OST baselines
> > > We implemented the OSST method of Zhang et al. using the authors' repo. The natural hyperparameter to tune in this setting is the regularization strength $\lambda \in [0, 1]$ which controls the tradeoff between the survival objective (IBS loss) and tree size; we swept this across 100 values $\\{0.01, 0.02, \ldots, 1\\}$ to match the other methods. The results for OSST (and ST for comparison) are below.
> > >
> > > ---
> > > |  | Metric | ST | OSST |
> > > |---|---|---|---|
> > > | **Synth-Nonlinear** | F1 | 0.28 (0.03) | 0.29 (0.01) |
> > > | | EPE | 0.66 (0.01) | 0.69 (0.00) |
> > > | | C-index | 0.59 (0.01) | 0.54 (0.00) |
> > > | | | |
> > > | **AIDS-CD4** |  EPE | 0.58 (0.10) | 0.54 (0.02) |
> > > |  |  Rej@10% | 0.08 (0.03) | 0.07 (0.01) |
> > > |  |  C-Index | 0.71 (0.05) | 0.71 (0.01) |
> > > |  |  Size | 0.32 (0.11) | 1.00 (0.00) |
> > > | | | |
> > > | **AIDS-Karnof** |  EPE | 0.65 (0.08) | 0.62 (0.03) |
> > > | |  Rej@10% | 0.11 (0.03) | 0.15 (0.01) |
> > > | |  C-Index | 0.64 (0.05) | 0.66 (0.03) |
> > > | |  Size | 0.21 (0.09) | 1.00 (0.00) |
> > > | | | |
> > > | **AIDS-prior** |  EPE | 0.69 (0.05) | 0.70 (0.00) |
> > > | |  Rej@10% | 0.06 (0.00) | 0.06 (0.00) |
> > > | |  C-Index | 0.56 (0.04) | 0.46 (0.02) |
> > > | |  Size | 0.11 (0.01) | 1.00 (0.00) |
> > > | | | |
> > > | **GBSG2-tsize** |  EPE | 0.68 (0.02) | 0.68 (0.00) |
> > > | |  Rej@10% | 0.04 (0.01) | 0.08 (0.01) |
> > > | |  C-Index | 0.58 (0.03) | 0.57 (0.01) |
> > > | |  Size | 0.26 (0.07) | 1.00 (0.00) |
> > > | | | |
> > > | **VLC-Karnof** |  EPE | 0.45 (0.10) | 0.57 (0.02) |
> > > | |  Rej@10% | 0.08 (0.03) | 0.04 (0.01) |
> > > | |  C-Index | 0.77 (0.07) | 0.70 (0.02) |
> > > | |  Size | 0.23 (0.05) | 1.00 (0.00) |
> > > | | | |
> > > | **WHAS-DBP** |  EPE | 0.65 (0.06) | 0.68 (0.01) |
> > > | |  Rej@10% | 0.21 (0.03) | 0.23 (0.02) |
> > > | |  C-Index | 0.68 (0.04) | 0.60 (0.02) |
> > > | |  Size | 0.15 (0.02) | 0.64 (0.08) |
> > > | | | |
> > > | **NASA** | EPE | 0.53 (0.01) | 0.65 (0.00) |
> > > | | C-Index | 0.73 (0.01) | 0.60 (0.00) |
> > > | | Precision | 1.00 (0.00) | 0.63 (0.01) |
> > > | | Recall | 0.40 (0.06) | 1.00 (0.00) |
> > > ---
> > >
> > > In 6/7 experiments, OSST selects the entire covariate space (Size = 1.00 or recall 1.00 in NASA). In the remaining case (WHAS-DBP), the EPE, rejection fraction, and C-index are worse than ST. The results are similarly poor on the synthetic data. OSST never approaches the leading methods, confirming our original prediction. Given these results, we do not expect that other OST variants would yield qualitatively different conclusions, since the optimal splitting objective is not aligned with the subgroup discovery objective (OSST's preference for the trivial subgroup is consistent with this mismatch, not an implementation artifact).
> > >
> > > ---
> > >
> > > ### DDGroup vs. CT
> > > We believe there is a misunderstanding as to the main claim of our paper. To clarify: our paper's primary contribution is the theoretical foundations and novel metrics (EPE, CRS) for subgroup discovery with the Cox model, along with seven novel algorithms — not just DDGroup. CT is also our method, and its strong performance on some datasets is a success of our framework, not a limitation. We agree that further empirical study is valuable, but defer this to future work given that the present paper + appendix is already four times the main text page limit.
> > >
> > > ---
> > >
> > > ### Higher dimensions
> > >
> > > We extended the synthetic data experiment to d=4 dimensions; here the Cox model itself *and* the subgroup definition are with respect to 4 covariates. We used the same number of datapoints ($n=2000$) as in the d=2 case, making this a more challenging setup. DDGroup is still able to obtain good results with an F1 score of 0.89 (slightly favoring precision over recall). DDG-CI (which uses a modification of the C-index for the rejection phase; this modification is also introduced by our work) also works well in this setting.
> > >
> > > | Method | F1 | EPE | C-Index |
> > > |---|---|---|---|
> > > | ST | 0.20 (0.02) | 0.67 (0.00) | 0.59 (0.01) |
> > > | Base | 0.29 (0.00) | 0.68 (0.00) | 0.59 (0.00) |
> > > | OSST | 0.29 (0.00) | 0.68 (0.00) | 0.59 (0.00) |
> > > | PRIM | 0.30 (0.00) | 0.67 (0.00) | 0.59 (0.00) |
> > > | CT | 0.60 (0.09) | 0.39 (0.07) | 0.81 (0.05) |
> > > | Random | 0.66 (0.04) | 0.54 (0.01) | 0.76 (0.01) |
> > > | DDG-PL | 0.72 (0.04) | 0.41 (0.03) | 0.85 (0.02) |
> > > | DDG-NE | 0.74 (0.02) | 0.40 (0.03) | 0.85 (0.01) |
> > > | DDG-CI | 0.88 (0.02) | 0.26 (0.02) | 0.92 (0.01) |
> > > | **DDGroup** | **0.89 (0.02)** | **0.26 (0.02)** | **0.92 (0.01)** |
> > >
> > > The runtimes did not increase significantly in this setting. Finally, we note that our focus on low-dimensional/univariate settings is not a contrivance of our paper; it is a very common setting in purely applied work. For a modern example, see [Sopik et al.](https://pmc.ncbi.nlm.nih.gov/articles/PMC12923670/), where many results are reported as unadjusted, *univariate* models. This is very common in clinical research: in many clinical papers the *primary* statistical conclusions are presented using univariate models. While considering higher-dimensional settings also has value, our contributions are already relevant to practical settings.

---

### Official Review · Reviewer_4jcR · 2026-03-16

**Soundness:** 3
**Presentation:** 3
**Significance:** 3
**Originality:** 3
**Overall Recommendation:** 4
**Confidence:** 3

**Summary:**

This paper addresses the challenge of finding interpretable subgroups in survival data where a Cox proportional hazards model fits well. The use Expected Prediction Entropy (EPE), which measures how confidently and correctly the model predicts pairwise failure orderings, and Conditional Rank Statistics (CRS), which evaluates whether individual points belong in a subgroup. Using these tools, they develop algorithms to select core subgroups, reject inconsistent points, and expand the subgroups while maintaining model reliability. Experiments on synthetic and real datasets show that this approach discovers subgroups with better model fit and interpretability than standard Cox models applied to the full population.

**Compliance With Llm Reviewing Policy:**

Affirmed.

**Key Questions For Authors:**

I believe the authors wisely addressed the reviewer's concerns of dependency between feature distribution and EPE by leaving it in the realm of future work. Would the authors be willing to explore the disentanglement of the two in this round of revisions?

**Limitations:**

No... just as the authors state: "There are many potential societal consequences of our work, none which we feel must be specifically highlighted here."

The authors can freely state how their specific design choices have societal impact without taking away from their technical novel contributions.

**Strengths And Weaknesses:**

Soundness:
The counterexamples motivating why the C-index and partial likelihood fail for subgroup discovery are convincing and well-constructed. The EPE and CRS are statistically well-grounded, and the 8-method comparison across synthetic and real data is thorough. However, the main theoretical guarantee assumes the algorithm already finds a good starting subgroup (Assumption A4), which is the hardest part of the problem.

Presentation:
Clearly written and well-structured. The NASA case study was a great use case to showcase tangibility.

Significance:
The problem is practically relevant for the showcased applications in medicine and engineering. The NASA case study demonstrates that discovered subgroups can reveal known physical phenomena and suggest actionable insights.

Originality:
The problem formulation and combination of EPE and CRS are novel. However, the EPE expression has appeared before as a training loss, and the algorithmic framework follows the existing DDGroup template rather than introducing a fundamentally new approach.

---

> ### Author Rebuttal · Authors · 2026-03-25
>
> Thank you for your thoughtful review. We address each of your concerns below.
>
> >Assumption A4 is very strong
>
> We agree that Assumption A4 is a strong assumption. However, we respectfully disagree that this is the hardest part of the problem, and we discuss the validity of Assumption A4 in depth in Appendix G. Specifically, Appendix G.1 shows why Assumption A4 is necessary, and Appendix G.2, Proposition G.1 gives sufficient conditions at the level of the data distribution where Assumption A4 will hold. Our experiments on synthetic data also show that A4 seems to hold in practice.
>
> As an additional note, we remark that an alternative version of Theorem 6.1 (the main performance guarantee) holds under a weaker version of A4. Specifically, if one assumes that the core group selection procedure returns a core group which is fully contained in *any* axis-aligned region R in which a Cox model is well-specified, and there is a similar effect size gap on the hazard function in and outside of R, then DDGroup will recover R given a large enough sample and effect size. The counterexample provided in Appendix G.1 means that this region R may not necessarily have the minimum EPE out of all candidate regions in the dataset, but nevertheless DDGroup will have found a large region with low EPE in which a Cox model fits the data. To summarize, one can make the following tradeoff:
>
> Change Assumption A4 to: core group is a subset of a region R in which the Cox model holds → DDGroup recovers the full region R, which has low EPE and a valid Cox model, but potentially not *minimal* EPE of all regions in the data.
>
> We have added this discussion to the paper.
>
> >Novelty of EPE/DDgroup
>
> We would like to emphasize that, although some components of our contributions have appeared in other forms in the literature, their novelty goes beyond simply combining existing components. Regarding the EPE, the fundamental properties which make it a suitable metric (Propositions 3.2 and 3.3) are novel. The extension of the DDGroup framework is also nontrivial; in fact, the authors of the original DDGroup paper list it as an important open problem (see the discussion section of that paper: https://arxiv.org/pdf/2305.00195). Finally, the CRS and its properties is, to the best of our knowledge, completely novel.
>
> >Dependency between feature distribution and EPE
>
> Based on our current understanding, there are two “levels” of covariate/metric disentanglement which one could attempt to reach. The easier one is to try to get the problem closer to the uniform covariate distribution that we studied theoretically. In this case, there is still a dependence between the covariates and the EPE, but it is clearly characterized by our Proposition 3.3 and can be handled accordingly. One possible way of achieving this would be to use weighted bootstrap sampling to make the covariates closer to uniform, then evaluating using the bootstrapped sample.
>
> The second “level” would be to develop an alternative metric which is completely independent of the covariate distribution. We spent a lot of time thinking about this question, but we conjecture that there may be some sense in which it is impossible to have a meaningful metric (i.e., which can actually distinguish between models) which is completely covariate insensitive. The intuition for this can be seen in the counterexample after Proposition 3.3 (pg. 4, right column): depending on the covariate distribution, it may be *fundamentally harder* to distinguish between patients. The most extreme example of this is a “subgroup” which consists of a single point {x} in feature space. According to the features, all units belonging to this subgroup are identical. In this case, the fact that the EPE in some sense automatically adapts to the underlying intrinsic difficulty of distinguishing between units may even be seen as a feature rather than a bug.
>
> >Limitations aren’t discussed
>
> We discuss technical limitations of our approach in Section 8 (Limitations & Future Work paragraph). The main potential negative societal impact is related to one of the technical limitations–namely, the need for valid p-values–, and we have added the following paragraph:
>
> “Our methods can be considered as data-driven hypothesis generation methods, and as such are susceptible to false positives, i.e., “discovering” a subgroup which does not really exist. If acted upon without independent validation, in a clinical setting, this could lead to patients being inappropriately included in or excluded from treatments in a clinical setting. We therefore emphasize that discovered subgroups should be treated as hypotheses for further investigation, not as definitive clinical guidance.”
>
> Thank you again for your constructive feedback. If you have any remaining questions, please do not hesitate to ask.

---

### Decision · Program_Chairs · 2026-04-30

**Decision:**

Accept (regular)

**Comment:**

All reviewers agree that finding subpopulations in data where the Cox model fits well compared to the overall population is an interesting and challenging problem. The paper is well written and structured, the theoretical analysis is thorough, the experiments on synthetic data show the method works in principle, and the case study shows the method works well in practice. What is a pity, is that the paper does not include practical results on health care data where the Cox model would be a natural fit, and especially, for which more than just a single variable is needed to select the subgroup; right now, a trivial baseline that simply evaluates all discrete variable values (eg. pre-descretized intervals) and returns the one with the best Cox-model fit could be a strong competitor. Similarly, the authors currently focus on returning a single subgroup only without discussing how to handle cases where multiple Cox-model admitting subgroups exist. Overall, I'm happy to recommend acceptance of the work, but I do urge the authors to address these points in the camera-ready copy.